# FAIR CLASSIFIERS WITHOUT FAIR TRAINING: AN INFLUENCE-GUIDED DATA SAMPLING APPROACH

## ABSTRACT

A fair classifier should ensure the benefit of people from different groups, while the group information is often sensitive and unsuitable for model training. Therefore, learning a fair classifier but excluding sensitive attributes in the training dataset is important. In this paper, we study learning fair classifiers without implementing fair training algorithms to avoid possible leakage of sensitive information. Our theoretical analyses validate the possibility of this approach, that traditional training on a dataset with an appropriate distribution shift can reduce both the upper bound for fairness disparity and model generalization error, indicating that fairness and accuracy can be improved simultaneously with simply traditional training. We then propose a tractable solution to progressively shift the original training data during training by sampling influential data, where the sensitive attribute of new data is not accessed in sampling or used in training. Extensive experiments on real-world data demonstrate the effectiveness of our proposed algorithm.

## 1 INTRODUCTION

Machine Learning (ML) has dramatically impacted numerous sectors nowadays, optimizing processes and decision-making across various domains, including credit scoring (Siddiqi, 2005), demand forecasting (Carbonneau et al., 2008), and healthcare (Zicari et al., 2021). Algorithmic fairness entails the principle that models should not exhibit biases toward specific protected groups, distinguished by characteristics such as race, gender, or disability. Fairness-aware learning has gained prominence to mitigate prediction disparities in classification tasks. In essence, the core idea of in-processing mitigation is to impose some fairness constraints during the training process (Donini et al., 2018; Hardt et al., 2016; Agarwal et al., 2018; Zafar et al., 2017; Wang et al., 2022), leading to a widely embraced paradigm in practice. However, there are two notable concerns. First, designing fairness constraints in most existing works heavily relies on the visibility of sensitive attributes in the training dataset, which are not always available due to privacy concerns. Second, enforcing fairness constraints or regularizers into machine learning models often leads to a tradeoff between fairness and accuracy (Kleinberg et al., 2016), underscoring a prevalent challenge: as we place greater emphasis on ensuring fairness, the predictive model performance might be weakened. With these two concerns in mind, our core inquiry rests upon the following question:

> *When sensitive attributes of training data are unknown and fair training algorithms are not employed, how can we develop a fair classifier without compromising the model accuracy?*

The core proposal of this paper is an influence-guided sampling strategy that actively samples new data to supplement the training set. Our solution is motivated by a set of theoretical findings presented in Section 3. Our analysis of the impact of data disparity on model fairness informs us that collecting new data to induce an appropriate distribution shift can effectively reduce fairness disparity without the need to implement any fair training algorithms on the updated training data.

This paper considers the following scenario: we are given a set of unlabeled candidate data (without sensitive attributes nor training labels) that we can solicit labels to supplement the training data (e.g., a candidate set of unlabeled images to solicit annotations to improve the fairness of an image classifier built on it). We desire a smart process that only samples and labels a small subset of the

most relevant data in a cost-efficient way such that applying a naive training process on this newly constructed dataset without imposing any fairness mitigation strategies would suffice to improve the model fairness substantially.

The core challenge in developing the sampling strategies is that we would have to predict the influence of a candidate sample on to model's fairness without accessing the sensitive attributes nor the training labels before the actual sampling happened. Limited by the absence of sensitive attributes, we identify influential examples that enhance fairness from an unlabeled dataset by leveraging an auxiliary validation set drawn from the test distribution. Specifically, we assess the impact of each example on fairness by comparing the gradient originating from a single data example with the gradient derived from the entire validation set. This comparison helps in quantifying the potential advantage of including this specific example with a proxy label in improving fairness. Intuitively, if the gradient obtained from a single data example has a similar direction to the gradient from the validation set, it indicates that incorporating this example contributes to enhancing the model's fairness. We name our solution Fair Influential Sampling (FIS).

The main contributions of our work are summarized as follows.

- Our theoretical analysis indicates that training on datasets with a strategically implemented distribution shift can effectively reduce both the upper bound for fairness disparity and model generalization error (Lemma 3.1, Theorem 3.2). This gives us a key insight that fairness and accuracy can be improved simultaneously even with simply traditional training. [Section 3]
- We introduce a tractable solution (Algorithm 1) that progressively shifts the original training data, wherein we sample influential examples from an unlabeled dataset based on the combined influences of prediction and fairness without ever accessing their sensitive attributes during the training process. [Section 4]
- Empirical experiments on real-world datasets (CelebA, COMPAS, Adult, and Jigsaw) substantiate our claims, indicating the effectiveness and potential of our proposed algorithm in achieving fairness for ML classifiers. [Section 5]

## 2 RELATED WORK

**Fair classification.** The fairness-aware learning methods, in general, can be categorized into pre-processing, in-processing, and post-processing methods. Pre-processing methods typically reweigh or distort the data examples to mitigate the identified biases (Asudeh et al., 2019; Azzalini et al., 2021; Tae & Whang, 2021; Sharma et al., 2020; Celis et al., 2020; Chawla et al., 2002; Zemel et al., 2013; Chen et al., 2018). More relevant to us is the *importance reweighting*, which assigns different weights to different training examples to ensure fairness across groups (Kamiran & Calders, 2012; Jiang & Nachum, 2020; Diesendruck et al., 2020; Choi et al., 2020; Qraitem et al., 2023; Li & Vasconcelos, 2019). Our proposed algorithm bears similarity to a specific case of importance reweighting, particularly the 0-1 reweighting applied to newly added data. The main advantage of our work, however, lies in its ability to operate without needing access to the sensitive attributes of either the new or training data. Other parallel studies utilize importance weighting to learn a complex fair generative model in the weakly supervised setting (Diesendruck et al., 2020; Choi et al., 2020), or to mitigate representation bias in training datasets (Li & Vasconcelos, 2019).

Post-processing methods typically enforce fairness on a learned model through calibration (Feldman, 2015; Feldman et al., 2015; Hardt et al., 2016). Although this approach is likely to decrease the disparity of the classifier, by decoupling the training from the fairness enforcement, this procedure may not lead to the best trade-off between fairness and accuracy (Woodworth et al., 2017; Pleiss et al., 2017). In contrast, our work can achieve a great tradeoff between fairness and accuracy because we reduce the fairness disparity by mitigating the adverse effects of distribution shifts on generalization error. Additionally, these post-processing techniques necessitate access to the sensitive attribute during the inference phase, which is often not available in many real-world scenarios.

**Accuracy-fairness tradeoff.** It has been demonstrated that there is an implicit trade-off between fairness and accuracy in the literature. Compared to the prior works (Menon & Williamson, 2018; Prost et al., 2019), our work does not require additional assumptions about the classifier and the characteristics of the training/testing datasets (for example, distribution shifts).

(Li & Liu, 2022) is a similar work that utilizes the influence function to reweight the data examples but requires re-training. Our method focuses on soliciting additional samples from an external unla-

beled dataset while (Li & Liu, 2022) reweights the existing and fixed training dataset. Our approach is more closely with a fair active learning approach (Anahideh et al., 2022). However, this fair active learning framework relies on sensitive attribute information while our algorithm does not.

**Distribution shifts.** Common research concerning distribution shifts necessitates extra assumptions to build theoretical connections between features and attributes, like causal graphs (Singh et al., 2021), correlation shifts (Roh et al., 2023), and demographic shifts (Giguere et al., 2022). In contrast, our approach refrains from making further assumptions about the properties of distribution shifts.

In this literature, many works have utilized distributionally robust optimization (DRO) to reduce fairness disparity without sensitive attribute information (Hashimoto et al., 2018; Kirichenko et al., 2022; Liu et al., 2021; Lahoti et al., 2020; Veldanda et al., 2023; Sohoni et al., 2020). Although these works also evaluate the worst-group performance in the context of fairness, their approach differs as they do not strive to equalize the loss across all groups. Besides, in these studies, accuracy and worst-case accuracy are used to showcase the efficacy of DRO. Essentially, they equate fairness with uniform accuracy across groups, implying a model is considered fair if it demonstrates equal accuracies for all groups. However, this specific definition of fairness is somewhat restrictive and does not align with more conventional fairness definitions like DP or EOD.

## 3 DECOUPLING FAIRNESS WITH DATA DISPARITY

In this section, we give a brief introduction to the problem settings and present a formal definition of fairness. Subsequently, we present the main theoretical results to understand the fairness-accuracy tradeoff, demonstrating data disparity's impact on fairness.

### 3.1 PRELIMINARIES

We consider two distributions, $\mathcal{P}$ (source or train) and $\mathcal{Q}$ (target or test), each defined as a probability distribution for certain *examples*, where each example defines values for three random variables: *feature $x$*; *label $y$*; and *sensitive attribute $a$*. Commonly, sensitive attributes $a$ are utilized for grouping purposes. Let $P$ ($Q$) denote the train (test) dataset sampled from the distribution $\mathcal{P}$ ($\mathcal{Q}$), where $P := \{(x_n, y_n, a_n)\}_{n \in N_P}$. Without loss of generality, we discrete the whole distribution space and suppose that the train/test distributions are both drawn from a series of component distributions $\{\pi_1, \cdots, \pi_I\}$ (Feldman, 2020).

Consider a typical optimization model, which can be defined as the following empirical risk $\mathcal{R}_P(\mathbf{w})$ over the train set $P$ by splitting samples based on the component distributions, shown as follows.

$$\mathcal{R}_P(\mathbf{w}) := \mathbb{E}_{(x,y)\in P}[\ell(f(x,\mathbf{w}),y)] = \sum_{i=1}^{I} p^{(P)}(\pi=i) \cdot \mathbb{E}_{(x,y)\sim\pi_i}[\ell(f(x,\mathbf{w}),y)]. \quad (1)$$

where $\mathbf{w}$ represents the model parameters to be optimized, example $\zeta = (x, y)$ uniformly sampled from dataset $\mathcal{D}_k$, $p^{(P)}(\pi=i)$ represents the frequencies that example in $P$ drawn from component distribution $\pi_i$, and $f(\cdot)$ indicate the classifier. Noting $l(\cdot, \cdot)$ represents the loss function, which can be an indicate function $\mathbb{I}(\cdot)$ to demonstrate 0-1 loss. Typically, it will apply $T$ epochs of SGD to obtain a converged model. We then introduce the definition of fairness disparity:

**Definition 3.1.** *(Fairness disparity (Hashimoto et al., 2018; Zafar et al., 2019)). Let $Q_k$ denote the partition of the dataset $Q$ for group $k$. Given the model parameters $\mathbf{w}^P$ trained on the dataset $P$, the fairness disparity between $Q$ and $Q_k$ is defined as: $\mathcal{R}_{Q_k}(\mathbf{w}^P) - \mathcal{R}_Q(\mathbf{w}^P)$, where $\mathcal{R}_Q(\mathbf{w}) := \mathbb{E}_{(x,y)\sim Q}[\ell(f(x,\mathbf{w}),y)]$ denotes the expected risk induced on the target distribution $\mathcal{Q}$.*

In fact, this definition has recently been highlighted in (Hashimoto et al., 2018; Zafar et al., 2019), which naturally quantifies the discrepancy in a trained model's performance between a specific subset and the entire test set. That is, a model can be deemed fair if it exhibits consistent performance for an individual group $Q_k$ as compared to the entire test set $Q$. For completeness, we also include additional well-known definitions of fairness:

**Definition 3.2.** *(Demographic Parity (DP)). A classifier $f$ adheres to demographic parity concerning the sensitive attribute $a$ if: $\mathbb{E}[f(x, \mathbf{w})] = \mathbb{E}[f(x, \mathbf{w})|a]$.*

**Definition 3.3.** *(Equalized Odds (EOd) (Hardt et al., 2016)). A classifier $f$ meets the equalized odds with respect to the sensitive attribute $a$ if: $\mathbb{E}[f(x, \mathbf{w})|y] = \mathbb{E}[f(x, \mathbf{w})|y, a]$.*

Even though there may be a general incompatibility between fairness disparity and popular fairness metrics such as DP or EOd, under the criteria of this fairness disparity definition, these metrics could be encouraged (Shui et al., 2022; Hashimoto et al., 2018).

**Remark 3.1** (Connections to other fairness definitions.) Definition 3.1 targets group-level fairness, which has similar granularity to the classical attribute-level fairness such as accuracy parity (Zafar et al., 2017), device-level parity (Li et al., 2019), or small prediction loss for groups (Zafar et al., 2019; Balashankar et al., 2019; Martinez et al., 2019; Hashimoto et al., 2018).

To reflect the unfairness problem implicitly hidden in models, we introduce two generalized basic assumptions in convergence analysis (Li et al., 2019).

**Assumption 3.1.** *(L-Lipschitz Continuous). There exists a constant $L > 0$, for any $\mathbf{v}, \mathbf{w} \in \mathbb{R}^d$, $\mathcal{R}_P(\mathbf{v}) \leq \mathcal{R}_P(\mathbf{w}) + \langle \nabla \mathcal{R}_P(\mathbf{w}), \mathbf{v} - \mathbf{w} \rangle + \frac{L}{2} \|\mathbf{v} - \mathbf{w}\|_2^2.$*

**Assumption 3.2.** *(Bounded Gradient on Random Sample). The stochastic gradients on any sample are uniformly bounded, i.e., $\mathbb{E}[||\nabla \mathcal{R}_P(\mathbf{w}, \zeta)||^2] \leq G^2$, and epoch $t \in [1, \cdots, T]$.*

## 3.2 Understanding fairness-accuracy tradeoff

Developing a fair classifier while maintaining model accuracy requires a comprehensive analysis of the intrinsic fairness-accuracy tradeoff. In general, this tradeoff is understood theoretically by analyzing the upper bound of generalization error and predefined fairness metrics (Shui et al., 2022; Huang & Vishnoi, 2019; Dutta et al., 2020; Wang et al., 2021). Therefore, as motivated, we initiate our discussion with a theoretical evaluation of the bounds of both generalization error and fairness disparity. We present the theoretical findings below.

For ease of presentation, we begin by establishing the measure of probability distance between two datasets or distributions, denoted as $\text{dist}(\mathcal{A}, \mathcal{B}) := \sum_{i=1}^{I} |p^{(\mathcal{A})}(\pi = i) - p^{(\mathcal{B})}(\pi = i)|$. Analogous to Assumption 3.2, we further make a mild assumption to bound the loss over the component distributions $\pi_i$ according corresponding model, that is, $\mathbb{E}_{(x,y) \sim \pi_i}[\ell(f(x, \mathbf{w}^P), y)] \leq G_P, \forall i \in I$.

**Lemma 3.1.** *(Generalization error bound). Let $\text{dist}(\mathcal{P}, \mathcal{Q})$, $G_P$ be defined therein. With probability at least $1 - \delta$ with $\delta \in (0, 1)$, the generalization error bound of the model trained on dataset $P$ is*

$$\mathcal{R}_{\mathcal{Q}}(\mathbf{w}^P) \leq \underbrace{G_P \cdot \text{dist}(\mathcal{P}, \mathcal{Q})}_{\text{distribution shift}} + \sqrt{\frac{\log(4/\delta)}{2N_P}} + \mathcal{R}_P(\mathbf{w}^P). \tag{2}$$

Note that the generalization error bound is predominantly influenced by the shift in distribution when we think of an overfitting model, i.e., the empirical risk $\mathcal{R}_P(\mathbf{w}^P) \to 0$.

**Theorem 3.2.** *(Upper bound of fairness disparity). Suppose $\mathcal{R}_{\mathcal{Q}}(\cdot)$ follows Assumption 3.1. Let $\text{dist}(\mathcal{P}, \mathcal{Q})$, $G_P$, $\text{dist}(\mathcal{P}_k, \mathcal{Q}_k)$ and $\text{dist}(P_k, P)$ be defined therein. Given model $\mathbf{w}^P$ and $\mathbf{w}^k$ trained exclusively on group $k$'s data $P_k$, with probability at least $1 - \delta$ with $\delta \in (0, 1)$, then the upper bound of the fairness disparity is*

$$\mathcal{R}_{\mathcal{Q}_k}(\mathbf{w}^P) - \mathcal{R}_{\mathcal{Q}}(\mathbf{w}^P) \leq \underbrace{G_P \cdot \text{dist}(\mathcal{P}, \mathcal{Q})}_{\text{distribution shift}} + G_k \cdot \text{dist}(\mathcal{P}_k, \mathcal{Q}_k) + \underbrace{\Phi \cdot \text{dist}(P_k, P)^2}_{\text{group gap}} + \Upsilon. \tag{3}$$

*where $\Phi = 4L^2 G^2 \sum_{t=0}^{T-1} (\eta_t^2 (1 + 2\eta_t^2 L^2))^t$, $\Upsilon = \sqrt{\frac{\log(4/\delta)}{2N_P}} + \sqrt{\frac{\log(4/\delta)}{2N_{P_k}}} + \varpi + \varpi_k$.*

*Note that $\mathbb{E}_{(x,y) \sim \pi_i}[\ell(f(x, \mathbf{w}^k), y)] \leq G_k$, $\varpi = \mathcal{R}_P(\mathbf{w}^P) - \mathcal{R}_{\mathcal{Q}}^*(\mathbf{w}^{\mathcal{Q}})$ and $\varpi_k = \mathcal{R}_{P_k}(\mathbf{w}^k) - \mathcal{R}_{\mathcal{Q}_k}^*(\mathbf{w}^{\mathcal{Q}_k})$. Specifically, $\varpi$ and $\varpi_k$ can be regarded as constants because $\mathcal{R}_P(\mathbf{w}^P)$ and $\mathcal{R}_{P_k}(\mathbf{w}^k)$ correspond to the empirical risks, $\mathcal{R}_{\mathcal{Q}}^*(\mathbf{w}^{\mathcal{Q}})$ and $\mathcal{R}_{\mathcal{Q}_k}^*(\mathbf{w}^{\mathcal{Q}_k})$ represent the ideal minimal empirical risk of model $\mathbf{w}^{\mathcal{Q}}$ trained on distribution $\mathcal{Q}$ and $\mathcal{Q}_k$, respectively.*

**Interpretations.** The upper bound in Eq. (3) illustrates several intuitive aspects that induced unfairness. (1) *Group biased data*. For group-level fairness, the more balanced the data is, the smaller the fairness disparity would be; (2) *distribution shift*. For source/target distribution, the closer the distribution is, the smaller the gap would be; (3) *Data size*. For training data size, the larger the size is (potentially eliminating data bias across groups), the smaller the gap would be.

**Main observation.** Note that Theorem 3.2 implies that the fairness disparity is essentially influenced by both the potential distribution shift and the inherent group disparity. In addition, Lemma 3.1

underscores how the generalization error is impacted by distribution shifts. Therefore, by integrating insights from Lemma 3.1, Theorem 3.2 clearly articulates a tradeoff between the fairness disparity and the generalization error. However, in practice, finding the optimal tradeoff between accuracy and fairness to reduce the fairness disparity directly is challenging due to the limited knowledge of the training and testing data distributions.

Therefore, the conventional methods that focus on mitigating the group gap may be not effective due to the adverse impact of the distribution shift. To elaborate, training on a shifted distribution $P$ may help alleviate the data disparity, but it can also introduce additional disparities, as indicated in the first term on the RHS of Theorem 3.2, leading to a performance drop. Our results provide guidance to reduce the fairness disparity by controlling the impact of the distribution shift through the generalization error, which corresponds to the practical model accuracy. That is, an appropriate distribution shift can reduce both the upper bound for fairness disparity and model generalization error. In an ideal scenario, the group gap in $P$ is smaller than the previous one in $Q$, and dist$(\mathcal{P}, \mathcal{Q})$ is also small. Therefore, in general, our goal is to find an appropriate distribution shift that enhances fairness without compromising on accuracy.

## 4 IMPROVING FAIRNESS WITH DATA INFLUENTIAL SAMPLING

Our theoretical results indicate a tradeoff between fairness disparity and the generalization error. The key observation is that the distribution shift typically with negative impacts on fairness can effectively be utilized to reduce the fairness disparity without sacrificing accuracy. As motivated, in this section, we propose an influence-guided sampling strategy that actively samples new data to supplement the training set such as mitigating fairness disparity.

### 4.1 FINDING INFLUENTIAL EXAMPLES

In this subsection, we introduce our basic scenario and then present key techniques used for sampling, which calculates the influence of prediction and fairness components.

#### 4.1.1 BASIC SCENARIO

Recall that our theoretical findings illustrate that collecting new data to induce an appropriate distribution shift can effectively reduce fairness disparity without the need to implement any fair training algorithms on the updated training data. To implement this, we consider the following basic scenario, involving a $K$-class classification task with a small training set denoted by $P := \{(x_n, y_n, a_n) | n \in [N_P]\}$, where $[N_P] := \{1, 2, \cdots, N_P\}$, $x_n$ denotes the feature vector, $y_n \in [K]$ denotes the label, and $a_n \in [M]$ denotes the sensitive attribute, respectively. We are given a large unlabeled dataset $U := \{(x_{n+N_P}, \cdot, \cdot) | n \in [N_U]\}$ that be used for training, where the labels and sensitive attributes are unknown but can be probed with a certain cost. In this context, the ultimate goal is to develop a model on both $P$ and a small set of $U$ while keeping the probing cost low and ensuring fairness constraints are met when evaluating its performance on the test dataset $Q := \{(x_n^\circ, y_n^\circ, a_n^\circ) | n \in [N_Q]\}$, drawing from data distribution $\mathcal{Q}$. Here, to tackle the absence of sensitive attributes for dataset $P$ and $U$, we utilize an auxiliary hold-out validation dataset $Q_v := \{(x_n^\circ, y_n^\circ, a_n^\circ) | n \in [N_v]\}$, which originates from $\mathcal{Q}$. Note that our access is limited to the sensitive attributes of the validation dataset; we do not even require a proxy for the sensitive attribute.

To explicitly measure fairness loss, similar to the prediction loss $\ell(f(x; \mathbf{w}), y)$, we define the fairness constraint as $\phi(\{f(x_n^\circ; \mathbf{w}), y_n^\circ, a_n^\circ | n \in [N]\})$. When there is no confusion, the empirical risk for a specific loss function shall be $\mathcal{R}_\ell = \sum_{n=1}^{N_P} \ell(f(x_n; \mathbf{w}), y_n)$. More specifically, the loss with one probed data can be formulated as follows.

$$\sum_{n=1}^{N_P} \ell(f(x_n; \mathbf{w}), y_n) + \underbrace{\ell(f(x'; \mathbf{w}), y')}_{\text{loss for the unlabeled example}} . \tag{4}$$

where label $y'$ is unknown before probing. It should be noted that before we solicit the true labels of samples from $U$ for training, $y'$ is used as the proxy label. In the following subsection 4.2, we will present two strategies for annotating the label $y'$.

### 4.1.2 CALCULATING INFLUENCE OF PREDICTION/FAIRNESS COMPONENT

Without any further information about train/test data distributions, a significant challenge arises in collecting a small number of data examples from the unlabeled set $U$ to mitigate fairness disparity without sacrificing accuracy. This becomes particularly intricate when there's no prior knowledge of the labels and attributes before probing.

Inspired by recent work (Diesendruck et al., 2020; Choi et al., 2020; Li & Vasconcelos, 2019), to address the lack of sensitive attributes, the high-level idea is that we can evaluate the impact of each example on fairness by comparing the gradient originating from a single data example with the gradient derived from the entire validation set. This comparison helps in quantifying the potential advantage of including this specific example in improving fairness. Intuitively, if the gradient obtained from a single data example has a similar direction to the gradient from the validation set, it indicates that incorporating this example contributes to enhancing the model's fairness. Similarly, the negative impact of a single data example on prediction can also be avoided by using its gradient.

In this regard, an ideal new example can help reduce the fairness disparity while keeping the accuracy non-decreasing. Here, we consider the loss for the prediction component and fairness component, i.e., $\ell(f(x; \mathbf{w}), y)$ and $\phi(f(x; \mathbf{w}), y, a)$, which can be differentiable. Supposed that the model is updated following gradient decent, the change of model parameters by counterfactually optimized on a new instance $(x', y')$ is

$$\mathbf{w}_{t+1} = \mathbf{w}_t - \eta \frac{\partial \ell(f(x'; \mathbf{w}), y')}{\partial \mathbf{w}}\bigg|_{\mathbf{w}=\mathbf{w}_t} \tag{5}$$

Following this, we compute the influence of the prediction and fairness components, respectively.

**Influence of Prediction Component.** The influence for the prediction component on the validation data $(x_n^\circ, y_n^\circ)$, when model parameters is updated from $\mathbf{w}_t$ to $\mathbf{w}_{t+1}$ by adding a new example $(x', y')$, is given by

$$\mathsf{Infl}_{\mathsf{acc}}((x_n^\circ, y_n^\circ), (x', y'); \mathbf{w}_t, \mathbf{w}_{t+1}) := \ell(f(x_n^\circ; \mathbf{w}_{t+1}), y_n^\circ) - \ell(f(x_n^\circ; \mathbf{w}_t), y_n^\circ).$$

For the ease of notation, we use $\mathsf{Infl}_{\mathsf{acc}}(n, x', y')$ to represent $\mathsf{Infl}_{\mathsf{acc}}((x_n^\circ, y_n^\circ), (x', y'); \mathbf{w}_t, \mathbf{w}_{t+1})$. By applying first-order Taylor expansion, we obtain the following closed-form statement:

**Lemma 4.1.** *The influence of predictions on the validation dataset $Q_v$ can be denoted by*

$$\mathsf{Infl}_{\mathsf{acc}}(x', y') := \sum_{n \in [N_v]} \mathsf{Infl}_{\mathsf{acc}}(n, x', y') \approx -\eta \left\langle \frac{\partial \ell(f(x'; \mathbf{w}), y')}{\partial \mathbf{w}}, \sum_{n=1}^{N_v} \left[ \frac{\partial \ell(f(x_n^\circ; \mathbf{w}), y_n^\circ)}{\partial \mathbf{w}} \right] \right\rangle \bigg|_{\mathbf{w}=\mathbf{w}_t}. \tag{6}$$

**Influence of Fairness Component.** Recall $\phi$ denotes the fairness loss and $[N_v]$ is the index set of validation data. The fairness influence on validation dataset $Q_v$ when model parameters change from $\mathbf{w}_t$ to $\mathbf{w}_{t+1}$ by adding $(x', y')$ is denoted by

$$\mathsf{Infl}_{\mathsf{fair}}(V, (x', y'); \mathbf{w}_t, \mathbf{w}_{t+1}) := \phi(\{f(x_n^\circ; \mathbf{w}_{t+1}), y_n^\circ, a_n^\circ | n \in [N_v]\}) - \phi(\{f(x_n^\circ; \mathbf{w}_t), y_n^\circ, a_n^\circ | n \in [N_v]\}). \tag{7}$$

For simplicity, we write $\mathsf{Infl}_{\mathsf{fair}}(V, (x', y'); \mathbf{w}_t, \mathbf{w}_{t+1})$ as $\mathsf{Infl}_{\mathsf{fair}}(x', y')$. Then, similarly, we have

**Lemma 4.2.** *The influence of fairness on the validation dataset $Q_v$ can be denoted by*

$$\mathsf{Infl}_{\mathsf{fair}}(x', y') \approx -\eta \sum_{n \in [N_v]} \left\langle \frac{\partial \ell(f(x'; \mathbf{w}), y')}{\partial \mathbf{w}}, \frac{\partial \phi(\{f(x_n^\circ; \mathbf{w}), y_n^\circ, a_n^\circ | n \in [N_v]\})}{\partial \mathbf{w}} \right\rangle \bigg|_{\mathbf{w}=\mathbf{w}_t}. \tag{8}$$

Note that it is straightforward to verify that neither the influence of prediction nor fairness components require the sensitive attributes of the sample $(x', y')$, which are unavailable in dataset $U$.

### 4.2 ALGORITHM: FAIR INFLUENTIAL SAMPLING (FIS)

**Label Annotation.** Drawing on Lemma 4.1 and 4.2, we can efficiently pinpoint those examples with the most negative fairness influence and negative prediction influence. This sampling method aids in reducing fairness disparity while sidestepping adverse impacts on accuracy. Before diving into presenting our algorithm, it is necessary to address the problem of not having access to true label

$y$. In practice, one can always recruit human annotators to get the ground-truth labels. However, we assume that it is still cost-ineffective and we can leverage the model trained on the set $P$ to generate the proxy labels. Nevertheless, we still need to acquire the true labels for the selected data examples. To tackle this problem, we propose to annotate the proxy labels with the model trained on the labeled set $P$. Specifically, we present two strategies for annotating the label $y'$ given $x'$.

**Strategy I**      **Use low-influence labels.** That is, $\hat{y} = \arg\min_{k \in [K]} |\mathsf{Infl}_{\mathsf{acc}}(x', k)|$, which corresponds to using the most uncertain point.

**Strategy II**      **Rely on model prediction.** That is, $\hat{y} = \arg\max_{k \in [K]} f(x; \mathbf{w})[k]$, where $f(x; \mathbf{w})[y]$ indicates the model's prediction probability on label $y$.

**Proposed algorithm.** Incorporating sampling and annotation techniques, we develop a Fair Influential Sampling (FIS) method to select a limited number of impactful examples from the extensive unlabeled set $U$. The full training algorithm for fair influential sampling is summarized in Algorithm 1. Note that the tolerance $\epsilon$ is applied to monitor the performance drop in validation accuracy. In Line 2, we initiate the process by training a classifier $f$ solely on dataset $P$, that is, performing a warm start. Subsequently, T-round sampling iterations of sampling are applied to amend the dataset $P$, aiming to reduce fairness disparity. Following the iterative fashion, FIS guesses labels using the proposed strategy I or II in Line 4. In particular, in Lines 5-6, we calculate prediction's and fairness's influences for examples using Lemma 4.1 and Lemma 4.2, respectively. Then, the true labels of top-$r$ samples with the most negative impact on fairness will be further inquired for further training. Note that although we propose two specific strategies for guessing labels, our algorithm is a flexible framework that is compatible with any other labeling method.

**Remark 1.** *Suppose that the model is trained with cross-entropy loss. The labels obtained through **Strategy II** are sufficient to minimize the influence of the prediction component, i.e., $\mathsf{Infl}_{\mathsf{acc}}(x', k)$. That said, the **Strategy II** will produce similar labels as **Strategy I**.*

---

**Algorithm 1** Fair influential sampling (FIS)

---

1: **Input:** train dataset $P$, unlabeled dataset $U$, validation dataset $Q_v$, $t = 0$, number of new examples in each round $r$, number of rounds $T$, tolerance $\epsilon$
    ### Training phrase: warm start ###
2: Train classifier $f$ solely on $P$ by minimizing the empirical risk $R_\ell$. Obtain model parameters $\mathbf{w}_1$ and validation accuracy (on $Q_v$) $\mathsf{VAL}_0$.
    ### Iterative sampling phrase ###
3: **for** $t$ **in** $\{1, 2, \cdots, T\}$ **do**
4:      Guess label $\hat{y}$ according to strategy I or II.
5:      Calculate the influence of prediction component by Eq. (6) and the influence of fairness component by Eq. (8).
6:      Get the scores of fair examples:

$$S_{\mathsf{o}} = \{\mathsf{Infl}_{\mathsf{fair}}(x, \hat{y}) \mid \mathsf{Infl}_{\mathsf{acc}}(x, \hat{y}) \leq 0, \mathsf{Infl}_{\mathsf{fair}}(x, \hat{y}) \leq 0\}$$

7:      $S_t = \{\}$
8:      **while** $|S_t| < r$ **do**
9:         Find top-$(r - |S_t|)$ samples (lowest fairness influence) and get the annotated samples:

$$\{(x'_n, y'_n)\} \leftarrow \mathsf{Top\text{-}}r(S_{\mathsf{o}})$$

10:         $S_t \leftarrow S_t \cup \{(x'_n, y'_n) \mid \mathsf{Infl}_{\mathsf{acc}}(x'_n, y'_n) \leq 0, \mathsf{Infl}_{\mathsf{fair}}(x'_n, y'_n) \leq 0\}$
11:      **end while**
    ### Training phrase: incorporate new data for training ###
12:      Train model $f$ on $P \cup S_1 \cdots \cup S_t$ with true inquired labels by minimizing $R_\ell$. Obtain model parameters $\mathbf{w}_{t+1}$ and validation accuracy (on $Q_v$) $\mathsf{VAL}_t - \epsilon$.
13: **end for**
    **Output:** models $\{\mathbf{w}_t \mid \mathsf{VAL}_t > \mathsf{VAL}_0\}$

---

# 5 EMPIRICAL RESULTS

In this section, we empirically show the disparate impact of groups and present the effectiveness of our proposed influential sampling method to mitigate fairness disparity.

## 5.1 EXPERIMENTAL SETUP

We evaluate the performance of our algorithm on four real-world datasets across different modalities: CelebA (Liu et al., 2015), Adult (Asuncion & Newman, 2007), COMPAS (Angwin et al., 2016), and Jigsaw (Jigsaw, 2018). We report results on three group fairness metrics: difference of demographic parity (DP), difference of equal odds (EOd), and difference of equality of opportunity (EOp).

We compare with three baselines: 1) Base: directly train the model solely on the labeled dataset $P$; 2) Random: train the model on the dataset $P$ and a subset of $Q$ by random sampling; 3) BALD (Branchaud-Charron et al., 2021): active sampling according to the mutual information; JTT-20 (Liu et al., 2021): reweighting those misclassified examples for retraining. Here, we examine a weight of 20 for misclassified examples in JTT. We present the results of FIS with two proposed labeling strategies: 1) FIS-Infl: train with low-influence labels and 2) FIS-Pred: train with model prediction. Note that we present the average result of the classifier $\mathbf{w}_t$ output from Algorithm 1.

## 5.2 PERFORMANCE RESULTS

**Results on image datasets.** First, we train a vision transformer with patch size $(8, 8)$ on the CelebA face attribute dataset (Liu et al., 2015). We select four binary classification targets including `Smile`, `Attractive`, `Young`, and `Big Nose`. Note that we initially allocate 2% of the data for training purposes and the remaining 98% for sampling. Then, we randomly select 10% of the test data for validation. For ease of computation, we only utilize the last two layers of the model to compute the influence of prediction and fairness. Table 1 and Table 2 both report the obtained accuracy and the corresponding values of fairness metrics. One main observation is that our proposed method FIS outperforms baselines with a significant margin on three fairness metrics while maintaining the same accuracy level. As mentioned by our observation in Theorem 3.2, this is because FIS scores examples priority based on the influence of fairness preferentially, and preventing potential accuracy reduction by checking the corresponding influence of prediction.

**Table 1:** We examine the performance of our methods on the **CelebA dataset**. The binary classification targets are `Smiling` and `Attractive`. We select `gender` as the sensitive attribute. We present the values of accuracy and three fairness metrics in the format: (`test accuracy`, `fairness metric`). In particular, we highlight the best values achieved for accuracy and fairness metrics in green and the worst in red.

| $\epsilon = 0.05$ | Smiling | | | Attractive | | |
|---|---|---|---|---|---|---|
| | dp | eop | eod | dp | eop | eod |
| Base | (0.847, 0.131) | (0.847, 0.061) | (0.847, 0.040) | (0.694, 0.395) | (0.707, 0.259) | (0.694, 0.280) |
| Random | (0.853, 0.132) | (0.863, 0.053) | (0.861, 0.031) | (0.696, 0.367) | (0.708, 0.253) | (0.696, 0.243) |
| BALD | (0.886, 0.150) | (0.886, 0.058) | (0.886, 0.030) | (0.734, 0.459) | (0.734, 0.299) | (0.734, 0.314) |
| JTT-20 | (0.883, 0.135) | (0.881, 0.054) | (0.881, 0.030) | (0.726, 0.418) | (0.726, 0.230) | (0.726, 0.269) |
| FIS-Infl | (0.877, 0.122) | (0.886, 0.040) | (0.882, 0.023) | (0.680, 0.285) | (0.695, 0.148) | (0.692, 0.148) |
| FIS-Pred | (0.880, 0.121) | (0.881, 0.046) | (0.880, 0.028) | (0.683, 0.290) | (0.696, 0.125) | (0.692, 0.131) |

**Results on tabular datasets.** Next, we work with multi-layer perceptron (MLP) trained on the Adult dataset (Asuncion & Newman, 2007) and Compas dataset (Angwin et al., 2016), respectively. The below detailed settings are the same for both two datasets. Note that we resample the datasets to balance the class and group membership (Chawla et al., 2002). The MLP model is a two-layer ReLU network with a hidden size of 64. Note that the dataset is randomly split into a training and a test set in a ratio of 80 to 20. Then, we randomly re-select 20% of the training set for initial training and the remaining 80% for sampling. Also, 20% examples of the test set are selected to form a validation set. We utilize the whole model to compute the prediction influence and fairness for examples.

Table 3 and Table 4 summarize the key results of the Adult and Compas datasets, respectively. Again, our algorithm outperforms baselines significantly on all fairness definitions. Observe that the results from FIS-Infl and FIS-Pred are identical. This is due to the labels in FIS-Infl ultimately becoming their corresponding model predictions in the pursuit of the lowest influence.

**Table 2:** We examine the performance of our methods on the **CelebA dataset**. The binary classification targets are `Young` and `Big Nose`. We select `gender` as the sensitive attribute. We present the values of accuracy and three fairness metrics in the format: (`test accuracy, fairness metric`). In particular, we highlight the best values achieved for accuracy and fairness metrics in green and the worst in red.

| $\epsilon = 0.05$ | Young | | | Big Nose | | |
|---|---|---|---|---|---|---|
| | dp | eop | eod | dp | eop | eod |
| Base | (0.753, 0.209) | (0.765, 0.097) | (0.753, 0.202) | (0.735, 0.221) | (0.744, 0.227) | (0.744, 0.209) |
| Random | (0.772, 0.152) | (0.762, 0.052) | (0.779, 0.173) | (0.770, 0.190) | (0.757, 0.218) | (0.765, 0.196) |
| BALD | (0.799, 0.187) | (0.797, 0.069) | (0.799, 0.186) | (0.790, 0.200) | (0.757, 0.225) | (0.769, 0.187) |
| JTT-20 | (0.803, 0.184) | (0.800, 0.075) | (0.796, 0.191) | (0.786, 0.232) | (0.787, 0.241) | (0.786, 0.209) |
| FIS-Infl | (0.766, 0.139) | (0.775, 0.043) | (0.769, 0.168) | (0.771, 0.156) | (0.765, 0.129) | (0.758, 0.155) |
| FIS-Pred | (0.769, 0.145) | (0.775, 0.043) | (0.769, 0.168) | (0.776, 0.120) | (0.764, 0.190) | (0.758, 0.155) |

**Table 3:** The performance of experiments conducted on the **Adult dataset**. The sensitive attributes are `sex` and `race`. We present the values of accuracy and the associated three fairness metrics in the format: (`test accuracy, fairness metric`). In particular, we highlight the best values achieved for accuracy and fairness metrics in green and the worst in red.

| $\epsilon = 0.05$ | income (sex) | | | income (age) | | |
|---|---|---|---|---|---|---|
| | dp | eop | eod | dp | eop | eod |
| Base | (0.668, 0.090) | (0.668, 0.058) | (0.668, 0.052) | (0.726, 0.283) | (0.726, 0.211) | (0.726, 0.204) |
| Random | (0.782, 0.071) | (0.773, 0.101) | (0.769, 0.069) | (0.797, 0.130) | (0.764, 0.161) | (0.791, 0.112) |
| BALD | (0.779, 0.075) | (0.771, 0.073) | (0.780, 0.040) | (0.797, 0.099) | (0.790, 0.126) | (0.790, 0.080) |
| JTT-20 | (0.695, 0.038) | (0.627, 0.043) | (0.695, 0.033) | (0.627, 0.051) | (0.695, 0.061) | (0.695, 0.054) |
| FIS-Infl | (0.779, 0.070) | (0.778, 0.055) | (0.776, 0.037) | (0.797, 0.106) | (0.794, 0.120) | (0.791, 0.080) |
| FIS-Pred | (0.779, 0.070) | (0.778, 0.055) | (0.776, 0.037) | (0.797, 0.106) | (0.794, 0.120) | (0.791, 0.080) |

**Table 4:** The performance of experiments conducted on the **COMPAS dataset (left)** and **Jigsaw dataset (right)**. The selected sensitive attribute is `race` for both two datasets. We present the values of accuracy and three fairness metrics in the format: (`test accuracy, fairness metric`). In particular, we highlight the best values achieved for accuracy and fairness metrics in green and the worst in red.

| $\epsilon = 0.05$ | recidivism | | | $\epsilon = 0.05$ | toxicity | | |
|---|---|---|---|---|---|---|---|
| | dp | eop | eod | | dp | eop | eod |
| Base | (0.669, 0.326) | (0.669, 0.254) | (0.669, 0.277) | Base | (0.612, 0.067) | (0.612, 0.015) | (0.612, 0.040) |
| Random | (0.686, 0.305) | (0.688, 0.261) | (0.686, 0.257) | Random | (0.684, 0.044) | (0.691, 0.025) | (0.678, 0.022) |
| BALD | (0.692, 0.329) | (0.687, 0.264) | (0.692, 0.264) | BALD | (0.782, 0.037) | (0.698, 0.037) | (0.713, 0.033) |
| JTT-20 | (0.641, 0.231) | (0.641, 0.187) | (0.641, 0.194) | JTT-20 | (0.722, 0.047) | (0.722, 0.024) | (0.719, 0.022) |
| FIS-Infl | (0.678, 0.302) | (0.691, 0.251) | (0.692, 0.253) | FIS-Infl | (0.786, 0.030) | (0.719, 0.014) | (0.722, 0.022) |
| FIS-Pred | (0.678, 0.302) | (0.691, 0.251) | (0.692, 0.253) | FIS-Pred | (0.786, 0.030) | (0.719, 0.014) | (0.722, 0.022) |

**Results on text datasets.** Lastly, we consider Jigsaw Comment Toxicity Classification (Jigsaw, 2018) with text data. We first encode each raw comment text into a 768-dimensional textual representation vector by using a pre-trained BERT (Devlin et al., 2018), and train an MLP with hidden size 256 to perform classification. Then, we randomly select 5% of the training set for initial training and the remaining 95% for sampling. Similarly, 20% examples of the test set are selected to form a validation set. The influences of prediction and fairness for examples are computed on the whole MLP model. Table 4 reports the key results of the Jigsaw dataset, illustrating the superiority of our algorithm, aligning with the observation of Theorem 3.2.

## 6    CONCLUSIONS

In this work, we explore the training of fair classifiers without using fairness-aware learning, aiming to prevent the potential exposure of sensitive attributes. Our theoretical findings confirm that by using traditional training on suitably shifted dataset distributions, we can decrease the bound of fairness disparity and model generalization error simultaneously. Motivated by the insights from our results, we propose a fair influential sampling method FIS to inquiry examples from a large unlabelled dataset to progressively shift the original training data during training, where the sensitive attribute of new examples is not accessed in sampling or used in training. Empirical experiments on real-world data validate the efficacy of our proposed method.

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

APPENDIX

# A MORE DETAILS OF RELATED WORK

**Accuracy-fairness tradeoff.** It has been demonstrated that there is an implicit trade-off between fairness and accuracy in the literature. Compared to the prior works, our work does not require additional assumptions about the classifier and the characteristics of the training/testing datasets (for example, distribution shifts). For example, Menon & Williamson (2018) is confined to binary classification scenarios, specifically Bayesian optimal classifiers. Additionally, those in-processing techniques (Menon & Williamson, 2018; Prost et al., 2019) attain this tradeoff through the use of fairness regularization, which necessarily requires the information of sensitive attributes. The primary benefit of our method is that it does not inquire about sensitive attribute information of training data, offering a more general framework for achieving an optimal accuracy-fairness tradeoff.

(Li & Liu, 2022) is a similar work that utilizes the influence function to reweight the data examples but requires re-training. Our approach focuses on incorporating additional samples from an external unlabeled dataset to mitigate disparity, aligning more closely with a fair active learning approach (Anahideh et al., 2022). However, the fair active learning framework relies on sensitive attribute information while our algorithm does not.

**Distribution shifts.** Common research concerning distribution shifts necessitates extra assumptions to build theoretical connections between features and attributes, like causal graphs (Singh et al., 2021), correlation shifts (Roh et al., 2023), and demographic shifts (Giguere et al., 2022). In contrast, our approach refrains from making further assumptions about the characteristics of distribution shifts. Instead, we directly utilize sampling methods to construct an appropriate distribution shift, aiming to achieve a balanced tradeoff between accuracy and fairness.

In this literature, many works have utilized distributionally robust optimization (DRO) to optimize the uncertainty. (Hashimoto et al., 2018) proposes an approach based on DRO to minimize the worst-case risk over all distributions close to the empirical distribution. The work (Kirichenko et al., 2022) mainly focuses on improving the worst-group accuracy of neural networks in the presence of spurious features. Liu et al. (2021) primarily focuses on the use of DRO to upweight the training set to improve group robustness. Similarly, (Lahoti et al., 2020) proposes an optimization approach, Adversarially Reweighted Learning, to improve worst-case performance over unobserved protected groups. In reality, (Veldanda et al., 2023) is an extension that specifically addresses the sensitivity of various hyper-parameters, such as the size of the validation set. (Sohoni et al., 2020) exploits these estimated subclasses by training a new model to optimize worst-case performance over all estimated subclasses using group distributionally robust optimization (GDRO). Although these works also evaluate the worst-group performance in the context of fairness, their approach differs as they do not strive to equalize the loss across all groups. Besides, in these studies, accuracy and worst-case accuracy are used to showcase the efficacy of DRO. Essentially, they equate fairness with uniform accuracy across groups, implying a model is considered fair if it demonstrates equal accuracies for all groups. However, this specific definition of fairness is somewhat restrictive and does not align with more conventional fairness definitions like DP or EOD.

# B OMITTED PROOFS

In this section, we present detailed proofs for the lemmas and theorems in Sections 3 and 4, respectively.

## B.1 PROOF OF LEMMA 3.1

**Lemma 3.1**. (Generalization error bound). Let $\text{dist}(\mathcal{P}, \mathcal{Q})$, $G_P$ be defined therein. With probability at least $1 - \delta$ with $\delta \in (0, 1)$, the generalization error bound of the model trained on dataset $P$ is

$$\mathcal{R}_{\mathcal{Q}}(\mathbf{w}^P) \leq \underbrace{G_P \cdot \text{dist}(\mathcal{P}, \mathcal{Q})}_{\text{distribution shift}} + \sqrt{\frac{\log(4/\delta)}{2N_P}} + \mathcal{R}_P(\mathbf{w}^P).$$

*Proof.* The generalization error bound is

$$\mathcal{R}_{\mathcal{Q}}(\mathbf{w}^P) = \underbrace{\left(\mathcal{R}_{\mathcal{Q}}(\mathbf{w}^P) - \mathcal{R}_{\mathcal{P}}(\mathbf{w}^P)\right)}_{\text{distribution shift}} + \underbrace{\left(\mathcal{R}_{\mathcal{P}}(\mathbf{w}^P) - \mathcal{R}_P(\mathbf{w}^P)\right)}_{\text{Hoeffding's inequality}} + \underbrace{\mathcal{R}_P(\mathbf{w}^P))}_{\text{empirical risk}}$$

$$\leq G_P \cdot \text{dist}(\mathcal{P}, \mathcal{Q}) + \sqrt{\frac{\log(4/\delta)}{2N_P}} + \mathcal{R}_P(\mathbf{w}^P)$$

For the first term (distribution shift), we have

$$
\begin{aligned}
&\mathcal{R}_{\mathcal{Q}}(\mathbf{w}^P) - \mathcal{R}_{\mathcal{P}}(\mathbf{w}^P) \\
&= \mathbb{E}_{(x,y)\sim\mathcal{Q}}[\ell(f(x,\mathbf{w}^P),y)] - \mathbb{E}_{(x,y)\sim\mathcal{P}}[\ell(f(x,\mathbf{w}^P),y)] \\
&= \sum_{i=1}^{I} p^{(\mathcal{Q})}(\pi=i)\mathbb{E}_{(x,y)\sim\pi_i}[\ell(f(x,\mathbf{w}^P),y)] - \sum_{i=1}^{I} p^{(\mathcal{P})}(\pi=i)\mathbb{E}_{(x,y)\sim\pi_i}[\ell(f(x,\mathbf{w}^P),y)] \\
&= \sum_{i=1}^{I} |p^{(\mathcal{P})}(\pi=i) - p^{(\mathcal{Q})}(\pi=i)|\mathbb{E}_{(x,y)\sim\pi_i}[\ell(f(x,\mathbf{w}^P),y)] \\
&\leq G_P \cdot \mathrm{dist}(\mathcal{P},\mathcal{Q}).
\end{aligned}
$$

where we define $\mathrm{dist}(\mathcal{P},\mathcal{Q}) = \sum_{i=1}^{I} |p^{(\mathcal{P})}(\pi=i) - p^{(\mathcal{Q})}(\pi=i)|$ and $\mathbb{E}_{(x,y)\sim\pi_i}[\ell(f(x,\mathbf{w}^P),y)] \leq G_P, \forall i \in I$ because of Assumption 3.2. To avoid misunderstanding, we use a subscript $P$ of the constant $G$ to clarify the corresponding model $\mathbf{w}^P$. Then, for the second term (Hoeffding inequality), with probability at least $1-\delta$, we have $|\mathcal{R}_{\mathcal{P}}(\mathbf{w}^P) - \mathcal{R}_P(\mathbf{w}^P)| \leq \sqrt{\frac{\log(4/\delta)}{2N_P}}$. $\qquad\square$

## B.2 PROOF OF THEOREM 3.2

**Theorem 3.2**. (Upper bound of fairness disparity). Suppose $\mathcal{R}_{\mathcal{Q}}(\cdot)$ follows Assumption 3.1. Let $\mathrm{dist}(\mathcal{P},\mathcal{Q})$, $G_P$, $\mathrm{dist}(\mathcal{P}_k,\mathcal{Q}_k)$ and $\mathrm{dist}(P_k,P)$ be defined therein. Given model $\mathbf{w}^P$ and $\mathbf{w}^k$ trained exclusively on group $k$'s data $P_k$, with probability at least $1-\delta$ with $\delta \in (0,1)$, then the upper bound of the fairness disparity is

$$
\mathcal{R}_{\mathcal{Q}_k}(\mathbf{w}^P) - \mathcal{R}_{\mathcal{Q}}(\mathbf{w}^P) \leq \underbrace{G_P \cdot \mathrm{dist}(\mathcal{P},\mathcal{Q})}_{\text{distribution shift}} + G_k \cdot \mathrm{dist}(\mathcal{P}_k,\mathcal{Q}_k) + \underbrace{\Phi \cdot \mathrm{dist}(P_k,P)^2}_{\text{group gap}} + \Upsilon.
$$

where

$$
\Phi = 4L^2 G^2 \sum_{t=0}^{T-1} (\eta_t^2(1+2\eta_t^2 L^2))^t, \quad \Upsilon = \sqrt{\frac{\log(4/\delta)}{2N_P}} + \sqrt{\frac{\log(4/\delta)}{2N_{P_k}}} + \varpi + \varpi_k
$$

Note that $\mathbb{E}_{(x,y)\sim\pi_i}[\ell(f(x,\mathbf{w}^k),y)] \leq G_k$, $\varpi = \mathcal{R}_P(\mathbf{w}^P) - \mathcal{R}_{\mathcal{Q}}^*(\mathbf{w}^{\mathcal{Q}})$ and $\varpi_k = \mathcal{R}_{P_k}(\mathbf{w}^k) - \mathcal{R}_{\mathcal{Q}_k}^*(\mathbf{w}^{\mathcal{Q}_k})$. Specifically, $\varpi$ and $\varpi_k$ can be regarded as constants because $\mathcal{R}_P(\mathbf{w}^P)$ and $\mathcal{R}_{P_k}(\mathbf{w}^k)$ correspond to the empirical risks, $\mathcal{R}_{\mathcal{Q}}^*(\mathbf{w}^{\mathcal{Q}})$ and $\mathcal{R}_{\mathcal{Q}_k}^*(\mathbf{w}^{\mathcal{Q}\mathbf{k}})$ represent the ideal minimal empirical risk of model $\mathbf{w}^{\mathcal{Q}}$ trained on distribution $\mathcal{Q}$ and $\mathcal{Q}_k$, respectively.

*Proof.* First of all, we have

$$
\begin{aligned}
\mathcal{R}_{\mathcal{Q}_k}(\mathbf{w}^P) - \mathcal{R}_{\mathcal{Q}}(\mathbf{w}^P) &= (\mathcal{R}_{\mathcal{Q}}(\mathbf{w}^{P_k}) - \mathcal{R}_{\mathcal{Q}}(\mathbf{w}^P)) + (\mathcal{R}_{\mathcal{Q}_k}(\mathbf{w}^P) - \mathcal{R}_{\mathcal{Q}}(\mathbf{w}^{P_k})) \\
&= (\mathcal{R}_{\mathcal{Q}}(\mathbf{w}^{P_k}) - \mathcal{R}_{\mathcal{Q}}(\mathbf{w}^P)) + (\mathcal{R}_{\mathcal{Q}_k}(\mathbf{w}^P) - \mathcal{R}_{\mathcal{Q}_k}(\mathbf{w}^{P_k})) + (\mathcal{R}_{\mathcal{Q}_k}(\mathbf{w}^{P_k}) - \mathcal{R}_{\mathcal{Q}}(\mathbf{w}^{P_k})) \\
&\leq (\mathcal{R}_{\mathcal{Q}}(\mathbf{w}^{P_k}) - \mathcal{R}_{\mathcal{Q}}(\mathbf{w}^P)) + (\mathcal{R}_{\mathcal{Q}_k}(\mathbf{w}^P) - \mathcal{R}_{\mathcal{Q}_k}(\mathbf{w}^{P_k}))
\end{aligned}
$$

where $\mathbf{w}^{P_k}$ represents the model trained exclusively on group $k$'s data. For simplicity, when there is no confusion, we use $\mathbf{w}^k$ to substitute $\mathbf{w}^{P_k}$. The inequality $\mathcal{R}_{\mathcal{Q}_k}(\mathbf{w}^k) - \mathcal{R}_{\mathcal{Q}}(\mathbf{w}^k) \leq 0$ holds due to the fact that the model tailored for a single group $k$ can not generalize well to the entirety of the test set $Q$.

Then, for the first term, we have

$$
\begin{aligned}
\mathcal{R}_{\mathcal{Q}}(\mathbf{w}^k) - \mathcal{R}_{\mathcal{Q}}(\mathbf{w}^P) &\overset{(a)}{\leq} \langle \nabla\mathcal{R}_{\mathcal{Q}}(\mathbf{w}^P), \mathbf{w}^k - \mathbf{w}^P \rangle + \frac{L}{2}\|\mathbf{w}^k - \mathbf{w}^P\|^2 \\
&\overset{(b)}{\leq} L\|\mathbf{w}^k - \mathbf{w}^P\|^2 + \frac{1}{2L}\|\nabla\mathcal{R}_{\mathcal{Q}}(\mathbf{w}^P)\|^2 \\
&\overset{(c)}{\leq} \underbrace{L\|\mathbf{w}^k - \mathbf{w}^P\|^2}_{\text{group gap}} + \underbrace{(\mathcal{R}_{\mathcal{Q}}(\mathbf{w}^P) - \mathcal{R}_{\mathcal{Q}}^*(\mathbf{w}^{\mathcal{Q}}))}_{\text{train-test model gap}}
\end{aligned}
$$

where inequality (a) holds because of the L-smoothness of expected loss $\mathcal{R}_{\mathcal{Q}}(\cdot)$, i.e., Assumption 3.1. Specifically, inequality (b) holds because, by Cauchy-Schwarz inequality and AM-GM inequality, we have

$$
\langle \nabla\mathcal{R}_{\mathcal{Q}}(\mathbf{w}^P), \mathbf{w}^k - \mathbf{w}^P \rangle \leq \frac{L}{2}\|\mathbf{w}^k - \mathbf{w}^P\|^2 + \frac{1}{2L}\|\nabla\mathcal{R}_{\mathcal{Q}}(\mathbf{w}^P)\|^2.
$$

Then, inequality (c) holds due to the L-smoothness of $\mathcal{R}_{\mathcal{Q}}(\cdot)$ (Assumption 3.1), we can get a variant of Polak-Łojasiewicz inequality, which follows

$$\|\nabla\mathcal{R}_{\mathcal{Q}}(\mathbf{w}^P)\|^2 \leq 2L(\mathcal{R}_{\mathcal{Q}}(\mathbf{w}^P) - \mathcal{R}_{\mathcal{Q}}^*(\mathbf{w}^{\mathcal{Q}})).$$

Following a similar idea, for the second term, we also have

$$\mathcal{R}_{\mathcal{Q}_k}(\mathbf{w}^P) - \mathcal{R}_{\mathcal{Q}_k}(\mathbf{w}^k) \leq L\|\mathbf{w}^P - \mathbf{w}^k\|^2 + (\mathcal{R}_{\mathcal{Q}_k}(\mathbf{w}^k) - \mathcal{R}_{\mathcal{Q}_k}^*(\mathbf{w}^{\mathcal{Q}_k}))$$

Combined with two terms, we have

$$\mathcal{R}_{\mathcal{Q}_k}(\mathbf{w}^P) - \mathcal{R}_{\mathcal{Q}}(\mathbf{w}^P) \leq \underbrace{(\mathcal{R}_{\mathcal{Q}}(\mathbf{w}^P) - \mathcal{R}_{\mathcal{Q}}^*(\mathbf{w}^{\mathcal{Q}}))}_{\text{train-test model gap}} + \underbrace{2L\|\mathbf{w}^k - \mathbf{w}^P\|^2}_{\text{group gap}} + (\mathcal{R}_{\mathcal{Q}_k}(\mathbf{w}^k) - \mathcal{R}_{\mathcal{Q}_k}^*(\mathbf{w}^{\mathcal{Q}_k}))$$

Lastly, integrating with Lemma B.1, B.2 and B.3, we can finish the proof. □

**Lemma B.1.** *(Train-test model gap) With probability at least $1 - \delta$, given the model $\mathbf{w}^P$ trained on train set P, we have*

$$\mathcal{R}_{\mathcal{Q}}(\mathbf{w}^P) - \mathcal{R}_{\mathcal{Q}}^*(\mathbf{w}^{\mathcal{Q}}) \leq G_P \cdot dist(\mathcal{P}, \mathcal{Q}) + \sqrt{\frac{\log(4/\delta)}{2N_P}} + \varpi.$$

*where $dist(\mathcal{P}, \mathcal{Q}) = \sum_{i=1}^{I} |p^{(\mathcal{P})}(\pi = i) - p^{(\mathcal{Q})}(\pi = i)|$ and $\mathbb{E}_{(x,y)\sim\pi_i}[\ell(f(x, \mathbf{w}^P), y)] \leq G_P, \forall i \in I$, and a constant $\varpi := \mathcal{R}_P(\mathbf{w}^P) - \mathcal{R}_{\mathcal{Q}}^*(\mathbf{w}^{\mathcal{Q}})$.*

*Proof.* First of all, we have,

$$\mathcal{R}_{\mathcal{Q}}(\mathbf{w}^P) - \mathcal{R}_{\mathcal{Q}}^*(\mathbf{w}^{\mathcal{Q}}) = \left(\mathcal{R}_{\mathcal{Q}}(\mathbf{w}^P) - \mathcal{R}_{\mathcal{P}}(\mathbf{w}^P)\right) + \mathcal{R}_{\mathcal{P}}(\mathbf{w}^P) - \mathcal{R}_{\mathcal{Q}}^*(\mathbf{w}^{\mathcal{Q}})$$

$$\leq G \cdot \text{dist}(\mathcal{P}, \mathcal{Q}) + \mathcal{R}_{\mathcal{P}}(\mathbf{w}^P) - \mathcal{R}_{\mathcal{Q}}^*(\mathbf{w}^{\mathcal{Q}})$$

$$\leq \underbrace{G \cdot \text{dist}(\mathcal{P}, \mathcal{Q})}_{\text{distribution shift}} + \underbrace{\left(\mathcal{R}_{\mathcal{P}}(\mathbf{w}^P) - \mathcal{R}_{P}(\mathbf{w}^P)\right)}_{\text{Hoeffding's inequality}} + \underbrace{\left(\mathcal{R}_{P}(\mathbf{w}^P) - \mathcal{R}_{\mathcal{Q}}^*(\mathbf{w}^{\mathcal{Q}})\right)}_{\text{overfitting \& ideal case}}$$

For the first term (distribution shift), we have

$$\mathcal{R}_{\mathcal{Q}}(\mathbf{w}^P) - \mathcal{R}_{\mathcal{P}}(\mathbf{w}^P)$$

$$= \mathbb{E}_{(x,y)\sim\mathcal{Q}}[\ell(f(x, \mathbf{w}^P), y)] - \mathbb{E}_{(x,y)\sim\mathcal{P}}[\ell(f(x, \mathbf{w}^P), y)]$$

$$= \sum_{i=1}^{I} p^{(\mathcal{Q})}(\pi = i)\mathbb{E}_{(x,y)\sim\pi_i}[\ell(f(x, \mathbf{w}^P), y)] - \sum_{i=1}^{I} p^{(\mathcal{P})}(\pi = i)\mathbb{E}_{(x,y)\sim\pi_i}[\ell(f(x, \mathbf{w}^P), y)]$$

$$\leq \sum_{i=1}^{I} |p^{(\mathcal{P})}(\pi = i) - p^{(\mathcal{Q})}(\pi = i)|\mathbb{E}_{(x,y)\sim\pi_i}[\ell(f(x, \mathbf{w}^P), y)]$$

$$\leq G_P \cdot \text{dist}(\mathcal{P}, \mathcal{Q}).$$

where we define $\text{dist}(\mathcal{P}, \mathcal{Q}) = \sum_{i=1}^{I} |p^{(\mathcal{P})}(\pi = i) - p^{(\mathcal{Q})}(\pi = i)|$ and $\mathbb{E}_{(x,y)\sim\pi_i}[\ell(f(x, \mathbf{w}^P), y)] \leq G_P, \forall i \in I$ because of Assumption 3.2. For the second term, with probability at least $1 - \delta$, we have $|\mathcal{R}_{\mathcal{P}}(\mathbf{w}^P) - \mathcal{R}_P(\mathbf{w}^P)| \leq \sqrt{\frac{\log(4/\delta)}{2N_P}}$. Note that the third term $\mathcal{R}_P(\mathbf{w}^P) - \mathcal{R}_{\mathcal{Q}}^*(\mathbf{w}^{\mathcal{Q}})$ can be regarded as a constant $\varpi$.because $\mathcal{R}_P(\mathbf{w}^P)$ is the empirical risk and $\mathcal{R}_{\mathcal{Q}}^*(\mathbf{w}^{\mathcal{Q}})$ is the ideal minimal empirical risk of model $\mathbf{w}^{\mathcal{Q}}$ trained on distribution $\mathcal{Q}$.

Therefore, with probability at least $1 - \delta$, given model $\mathbf{w}^P$,

$$\mathcal{R}_{\mathcal{Q}}(\mathbf{w}^P) - \mathcal{R}_{\mathcal{Q}}^*(\mathbf{w}^{\mathcal{Q}}) \leq G_P \cdot \text{dist}(\mathcal{P}, \mathcal{Q}) + \sqrt{\frac{\log(4/\delta)}{2N_P}} + \varpi.$$

□

**Lemma B.2.** *(Group gap) Suppose Assumptions 3.1 and 3.2 hold for empirical risk $\mathcal{R}_P(\cdot)$, then we have*

$$\|\mathbf{w}^k - \mathbf{w}^P\|^2 \leq 2LG^2 \sum_{t=0}^{T}(\eta_t^2(1 + 2\eta_t^2 L^2))^t \left(\sum_{i=1}^{I}\left|p^{(k)}(\pi = i) - p^{(P)}(\pi = i)\right|\right)^2.$$

*where $\eta_t$ is epoch t's learning rate and T is the number of epochs.*

*Proof.* According to the above definition, we similarly define the following empirical risk $\mathcal{R}_{P_k}(\mathbf{w})$ over group $k$'s data $P_k$ by splitting samples according to their marginal distributions, shown as follows.

$$\mathcal{R}_{P_k}(\mathbf{w}) := \sum_{i=1}^{I} p^{(k)}(\pi = i)\mathbb{E}_{(x,y)\sim\pi_i}[\ell(f(x, \mathbf{w}), y)].$$

Let $\eta_t$ indicate the learning rate of epoch $t$. Then, for each epoch $t$, group $k$'s optimizer performs SGD as the following:

$$\mathbf{w}_t^k = \mathbf{w}_{t-1}^k - \eta_t \sum_{i=1}^{I} p^{(k)}(\pi = i)\nabla_{\mathbf{w}}\mathbb{E}_{(x,y)\sim\pi_i}[\ell(f(x, \mathbf{w}_{t-1}^k), y)].$$

For any epoch $t + 1$, we have

$\|\mathbf{w}_{t+1}^k - \mathbf{w}_{t+1}^P\|^2$

$= \|\mathbf{w}_t^k - \eta_t \sum_{i=1}^{I} p^{(k)}(\pi = i)\nabla_{\mathbf{w}}\mathbb{E}_{(x,y)\sim\pi_i}[\ell(f(x, \mathbf{w}_t^k), y)] - \mathbf{w}_t^P + \eta_t \sum_{i=1}^{I} p^{(P)}(\pi = i)\nabla_{\mathbf{w}}\mathbb{E}_{(x,y)\sim\pi_i}[\ell(f(x, \mathbf{w}_t^P), y)]\|^2$

$\leq \|\mathbf{w}_t^k - \mathbf{w}_t^P\|^2 + \eta_t^2\|\sum_{i=1}^{I} p^{(k)}(\pi = i)\nabla_{\mathbf{w}}\mathbb{E}_{(x,y)\sim\pi_i}[\ell(f(x, \mathbf{w}_t^k), y)] - \sum_{i=1}^{I} p^{(P)}(\pi = i)\nabla_{\mathbf{w}}\mathbb{E}_{(x,y)\sim\pi_i}[\ell(f(x, \mathbf{w}_t^P), y)]\|^2$

$\leq \|\mathbf{w}_t^k - \mathbf{w}_t^P\|^2 + 2\eta_t^2\|\sum_{i=1}^{I} p^{(P)}(\pi = i)L_{\pi_i}\left[\nabla_{\mathbf{w}}\mathbb{E}_{(x,y)\sim\pi_i}[\ell(f(x, \mathbf{w}_t^k), y)] - \nabla_{\mathbf{w}}\mathbb{E}_{(x,y)\sim\pi_i}[\ell(f(x, \mathbf{w}_t^P), y)]\right]\|^2$

$\quad + 2\eta_t^2\|\sum_{i=1}^{I} \left(p^{(k)}(\pi = i) - p^{(P)}(\pi = i)\right)\nabla_{\mathbf{w}}\mathbb{E}_{(x,y)\sim\pi_i}[\ell(f(x, \mathbf{w}_t^P), y)]\|^2$

$\leq \|\mathbf{w}_t^k - \mathbf{w}_t^P\|^2 + 2\eta_t^2\left(\sum_{i=1}^{I} p^{(k)}(\pi = i)L_{\pi_i}\right)^2\|\mathbf{w}_t^k - \mathbf{w}_t^P\|^2$

$\quad + 2L\eta_t^2 g_{max}^2(\mathbf{w}_t^Q)\left(\sum_{i=1}^{I} |p^{(k)}(\pi = i) - p^{(P)}(\pi = i)|\right)^2$

$\leq \left(1 + 2\eta_t^2\left(\sum_{i=1}^{I} p^{(k)}(\pi = i)L_{\pi_i}\right)^2\right)\|\mathbf{w}_t^k - \mathbf{w}_t^P\|^2$

$\quad + 2L\eta_t^2 g_{max}^2(\mathbf{w}_t^Q)\left(\sum_{i=1}^{I} |p^{(k)}(\pi = i) - p^{(P)}(\pi = i)|\right)^2$

$\leq (1 + 2\eta_t^2 L^2)\|\mathbf{w}_t^k - \mathbf{w}_t^P\|^2 + 2L\eta_t^2 G^2\left(\sum_{i=1}^{I} |p^{(k)}(\pi = i) - p^{(P)}(\pi = i)|\right)^2.$

where the third inequality holds since we assume that $\nabla_{\mathbf{w}}\mathbb{E}_{(x,y)\sim\pi_i}[\ell(f(x, \mathbf{w}), y)]$ is $L_{\pi_i}$-Lipschitz continuous, *i.e.*, $\|\nabla_{\mathbf{w}}\mathbb{E}_{(x,y)\sim\pi_i}[\ell(f(x, \mathbf{w}_t^k), y)] - \nabla_{\mathbf{w}}\mathbb{E}_{(x,y)\sim\pi_i}[\ell(f(x, \mathbf{w}_t^P), y)]\| \leq L_{\pi_i}\|\mathbf{w}_t^k - \mathbf{w}_t^P\|$, and denote $g_{max}(\mathbf{w}_t^i) = \max_{i=1}^{I}\|\nabla_{\mathbf{w}}\mathbb{E}_{(x,y)\sim\pi_i}[\ell(f(x, \mathbf{w}_t^P), y)]\|$. The last inequality holds because the above-mentioned assumption that $L = L_{\pi_i} = L_\pi, \forall i \in I$, *i.e.*, Lipschitz-continuity will not be affected by the samples' classes. Then, $g_{max}(\mathbf{w}_t^P) \leq G$ because of Assumption 3.2.

For $T$ training epochs, we have

$\|\mathbf{w}_T^k - \mathbf{w}_T^P\|^2$

$\leq (1 + 2\eta_t^2 L^2)\|\mathbf{w}_{T-1}^k - \mathbf{w}_{T-1}^P\|^2 + 2L\eta_t^2 G^2\left(\sum_{i=1}^{I} |p^{(k)}(\pi = i) - p^{(P)}(\pi = i)|\right)^2$

$\leq \prod_{t=0}^{T}(1 + 2\eta_t^2 L^2)^t\|\mathbf{w}_0^k - \mathbf{w}_0^P\|^2 + 2LG^2\sum_{t=0}^{T}(\eta_t^2(1 + 2\eta_t^2 L^2))^t\left(\sum_{i=1}^{I} |p^{(k)}(\pi = i) - p^{(P)}(\pi = i)|\right)^2$

$\leq 2LG^2\sum_{t=0}^{T}(\eta_t^2(1 + 2\eta_t^2 L^2))^t\left(\sum_{i=1}^{I} |p^{(k)}(\pi = i) - p^{(P)}(\pi = i)|\right)^2.$

where the last inequality holds because the initial models are the same, i.e., $\mathbf{w}_0 = \mathbf{w}_0^k = \mathbf{w}_0^P, \forall k$. $\qquad\square$

**Lemma B.3.** *With probability at least $1 - \delta$, given the model $\mathbf{w}^k$ trained on group $k$'s dataset $P_k$, we have*

$$\mathcal{R}_{\mathcal{Q}_k}(\mathbf{w}^k) - \mathcal{R}^*_{\mathcal{Q}_k}(\mathbf{w}^{\mathcal{Q}_k}) \leq G_k \cdot dist(\mathcal{P}_k, \mathcal{Q}_k) + \sqrt{\frac{\log(4/\delta)}{2N_{P_k}}} + \varpi_k.$$

*where $dist(\mathcal{P}_k, \mathcal{Q}_k) = \sum_{i=1}^I |p^{(\mathcal{P}_k)}(\pi = i) - p^{(\mathcal{Q}_k)}(\pi = i)|$ and $\mathbb{E}_{(x,y)\sim\pi_i}[\ell(f(x, \mathbf{w}^k), y)] \leq G_k, \forall i \in I$, and $\varpi_k := \mathcal{R}_{P_k}(\mathbf{w}^k) - \mathcal{R}^*_{\mathcal{Q}_k}(\mathbf{w}^{\mathcal{Q}_k})$.*

*Proof.* Building upon the proof idea presented in Lemma B.1, for completeness, we provide a full proof here. Firstly, we have,

$$\begin{aligned}
&\mathcal{R}_{\mathcal{Q}_k}(\mathbf{w}^k) - \mathcal{R}^*_{\mathcal{Q}_k}(\mathbf{w}^{\mathcal{Q}_k}) \\
&= \underbrace{(\mathcal{R}_{\mathcal{Q}_k}(\mathbf{w}^k) - \mathcal{R}_{\mathcal{P}_k}(\mathbf{w}^k))}_{\text{distribution shift}} + \underbrace{(\mathcal{R}_{\mathcal{P}_k}(\mathbf{w}^k) - \mathcal{R}_{P_k}(\mathbf{w}^k))}_{\text{Hoeffding's inequality}} + \underbrace{(\mathcal{R}_{P_k}(\mathbf{w}^k) - \mathcal{R}^*_{\mathcal{Q}_k}(\mathbf{w}^{\mathcal{Q}_k}))}_{\text{overfitting \& ideal case}}
\end{aligned}$$

For the first term, we have

$$\begin{aligned}
&\mathcal{R}_{\mathcal{Q}_k}(\mathbf{w}^k) - \mathcal{R}_{\mathcal{P}_k}(\mathbf{w}^k) \\
&= \sum_{i=1}^I p^{(\mathcal{Q}_k)}(\pi = i)\mathbb{E}_{(x,y)\sim\pi_i}[\ell(f(x, \mathbf{w}^k), y)] - \sum_{i=1}^I p^{(\mathcal{P}_k)}(\pi = i)\mathbb{E}_{(x,y)\sim\pi_i}[\ell(f(x, \mathbf{w}^k), y)] \\
&\leq \sum_{i=1}^I |p^{(\mathcal{P}_k)}(\pi = i) - p^{(\mathcal{Q}_k)}(\pi = i)|\mathbb{E}_{(x,y)\sim\pi_i}[\ell(f(x, \mathbf{w}^k), y)] \\
&\leq G_k \cdot \text{dist}(\mathcal{P}_k, \mathcal{Q}_k).
\end{aligned}$$

where $\text{dist}(\mathcal{P}_k, \mathcal{Q}_k) := \sum_{i=1}^I |p^{(\mathcal{P}_k)}(\pi = i) - p^{(\mathcal{Q}_k)}(\pi = i)|$ and $\mathbb{E}_{(x,y)\sim\pi_i}[\ell(f(x, \mathbf{w}^k), y)] \leq G_k, \forall i \in I$ due to Assumption 3.2. Recall that the constant $G_k$ clarifies the bound of loss on the corresponding model $\mathbf{w}^k$. For the second term, with probability at least $1 - \delta$, we have $|\mathcal{R}_{\mathcal{P}_k}(\mathbf{w}^k) - \mathcal{R}_{P_k}(\mathbf{w}^k)| \leq \sqrt{\frac{\log(4/\delta)}{2N_{P_k}}}$. For the third term, we define $\varpi_k := \mathcal{R}_{P_k}(\mathbf{w}^k) - \mathcal{R}^*_{\mathcal{Q}_k}(\mathbf{w}^{\mathcal{Q}_k})$, which can be regarded as a constant. This is because $\mathcal{R}_{P_k}(\mathbf{w}^k)$ represents empirical risk and $\mathcal{R}^*_{\mathcal{Q}_k}(\mathbf{w}^{\mathcal{Q}_k})$ is the ideal minimal empirical risk of model $\mathbf{w}^{\mathcal{Q}_k}$ trained on sub-distribution $\mathcal{Q}_k$.

Therefore, with probability at least $1 - \delta$, given model $\mathbf{w}^k$,

$$\mathcal{R}_{\mathcal{Q}_k}(\mathbf{w}^k) - \mathcal{R}^*_{\mathcal{Q}_k}(\mathbf{w}^{\mathcal{Q}_k}) \leq G_k \cdot \text{dist}(\mathcal{P}_k, \mathcal{Q}_k) + \sqrt{\frac{\log(4/\delta)}{2N_{P_k}}} + \varpi_k.$$

$\square$

### B.3 PROOF OF LEMMA 4.1

*Proof.* Taking the first-order Taylor expansion, we will have

$$\ell(f(x_n^\circ; \mathbf{w}_{t+1}), y_n^\circ) \approx \ell(f(x_n^\circ; \mathbf{w}_t), y_n^\circ) + \left\langle \left.\frac{\partial\ell(f(x_n^\circ; \mathbf{w}), y_n^\circ)}{\partial f(x_n^\circ; \mathbf{w})}\right|_{\mathbf{w}=\mathbf{w}_t}, f(x_n^\circ; \mathbf{w}_{t+1}) - f(x_n^\circ; \mathbf{w}_t) \right\rangle.$$

Similarly, we have

$$f(x_n^\circ; \mathbf{w}_{t+1}) - f(x_n^\circ; \mathbf{w}_t) \approx -\eta \left\langle \frac{\partial f(x_n^\circ; \mathbf{w})}{\partial \mathbf{w}}, \frac{\partial\ell(f(x'; \mathbf{w}), y')}{\partial \mathbf{w}} \right\rangle \Bigg|_{\mathbf{w}=\mathbf{w}_t}.$$

Therefore,

$$\ell(f(x_n^\circ; \mathbf{w}_{t+1}), y_n^\circ) - \ell(f(x_n^\circ; \mathbf{w}_t), y_n^\circ) \approx -\eta \left\langle \frac{\partial\ell(f(x'; \mathbf{w}), y')}{\partial \mathbf{w}}, \frac{\partial\ell(f(x_n^\circ; \mathbf{w}), y_n^\circ)}{\partial \mathbf{w}} \right\rangle \Bigg|_{\mathbf{w}=\mathbf{w}_t}.$$

Then the accuracy influence on the validation dataset $V$ can be denoted by

$$\text{Infl}_{\text{acc}}(x', y') := \sum_{n \in [N_V]} \text{Infl}_{\text{acc}}(n, x', y') \approx -\eta \left\langle \frac{\partial\ell(f(x'; \mathbf{w}), y')}{\partial \mathbf{w}}, \sum_{n=1}^{N_V} \left[\frac{\partial\ell(f(x_n^\circ; \mathbf{w}), y_n^\circ)}{\partial \mathbf{w}}\right] \right\rangle \Bigg|_{\mathbf{w}=\mathbf{w}_t}.$$

$\square$

### B.4 Proof of Lemma 4.2

*Proof.* By first-order approximation, we have

$$\phi(\{f(x_n^\circ; \mathbf{w}_{t+1}), y_n^\circ, a_n^\circ | n \in [N_V]\}) \approx \phi(\{f(x_n^\circ; \mathbf{w}_t), y_n^\circ, a_n^\circ | n \in [N_V]\})$$

$$+ \sum_{n \in [N_V]} \left\langle \frac{\partial \phi(\{f(x_n^\circ; \mathbf{w}), y_n^\circ, a_n^\circ | n \in [N_V]\})}{\partial f(x_n^\circ; \mathbf{w})} \bigg|_{\mathbf{w} = \mathbf{w}_t}, f(x_n^\circ; \mathbf{w}_{t+1}) - f(x_n^\circ; \mathbf{w}_t) \right\rangle.$$

Recall by first-order approximation, we have

$$f(x_n^\circ; \mathbf{w}_{t+1}) - f(x_n^\circ; \mathbf{w}_t) \approx -\eta \left\langle \frac{\partial f(x_n^\circ; \mathbf{w})}{\partial \mathbf{w}}, \frac{\partial \ell(f(x'; \mathbf{w}), y')}{\partial \mathbf{w}} \right\rangle \bigg|_{\mathbf{w} = \mathbf{w}_t}.$$

Note the loss function in the above equation should be $\ell$ since the model is updated with $\ell$-loss. Therefore,

$$\mathsf{Infl}_{\mathsf{fair}}(x', y') \approx -\eta \sum_{n \in [N_V]} \left\langle \frac{\partial \ell(f(x'; \mathbf{w}), y')}{\partial \mathbf{w}}, \frac{\partial \phi(\{f(x_n^\circ; \mathbf{w}), y_n^\circ, a_n^\circ | n \in [N_V]\})}{\partial \mathbf{w}} \right\rangle \bigg|_{\mathbf{w} = \mathbf{w}_t}.$$

□

## C Additional Experimental Result

**Exploring the impact of Label Budgets.** In our study, we examine how varying label budgets $r$ influence the balance between accuracy and fairness. We present the results of test accuracy and fairness disparity across different label budgets on the CelebA, Compas, and Jigsaw datasets. In these experiments, we use the demographics parity (DP) as our fairness metric. For convenience, we maintain a fixed label budget per round, using rounds of label budget allocation to demonstrate its impact. The designated label budgets per round for the CelebA, Compas, and Jigsaw are 256, 128, and 512, respectively. In the following figures, the $x$-axis is both the number of label budget rounds. The $y$-axis for the left and right sub-figures are test accuracy and DP gap, respectively. As observed in Figures 1-3, compared to the BALD baseline, our approach substantially reduces the DP gap without sacrificing test accuracy.

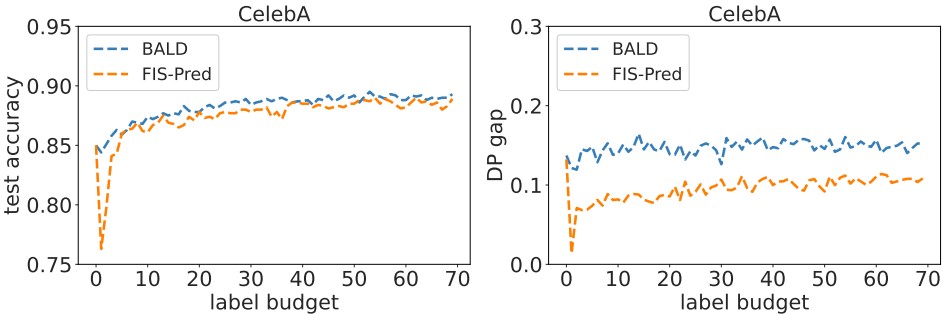

Figure 1: The impact of label budgets on the test accuracy & DP gap.

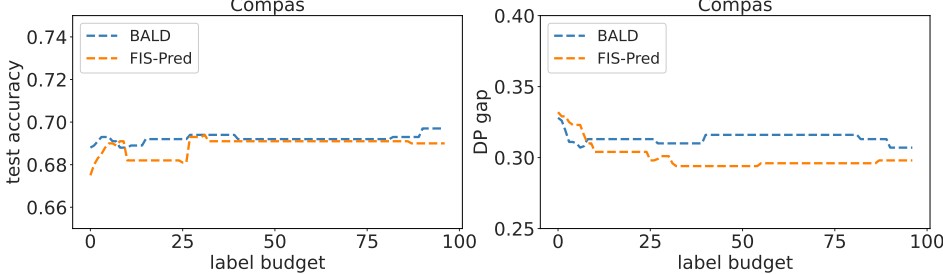

Figure 2: The impact of label budgets on the test accuracy & DP gap.

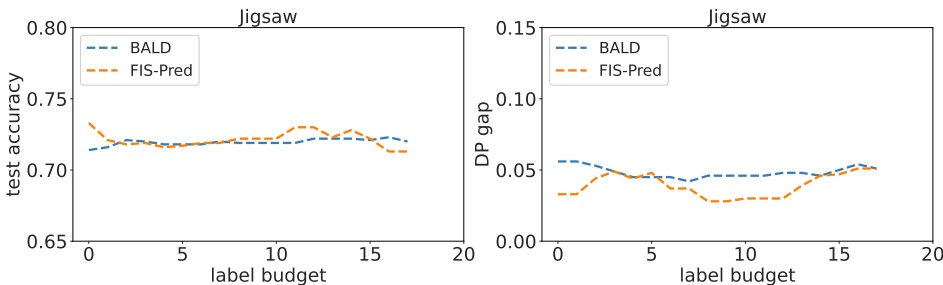

Figure 3: The impact of label budgets on the test accuracy & DP gap.

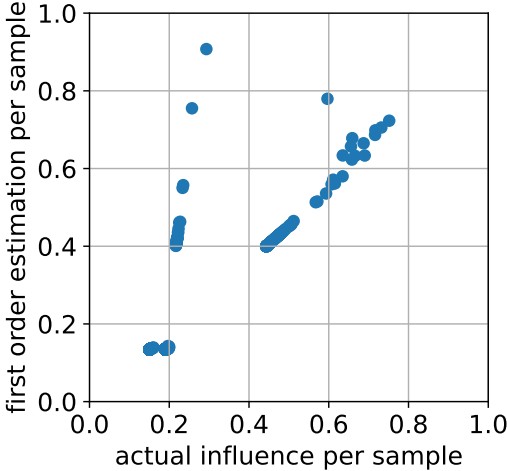

Figure 4: We validate how accurate the first-order estimation of the influence is in comparison to the real influence. The $x$-axis represents the actual influence per sample, and the $y$-axis represents the estimated influence. We observe that while some of the examples are away from the diagonal line (which indicates the estimation is inaccurate), the estimated influence for most of the data samples are very close to their actual influence values.

