# OpenReview forum: "Fair Classifiers Without Fair Training: An Influence-Guided Data Sampling Approach"
_ICLR.cc/2024/Conference — Submitted to ICLR 2024_

### Official Review · Reviewer_6fEn · 2023-10-30

**Soundness:** 2 fair
**Presentation:** 2 fair
**Contribution:** 2 fair
**Rating:** 5
**Confidence:** 2

**Summary:**

In fair classification, researchers tend to intervene during pre-processing, training or post-processing. The paper examines how to develop a fair classifier when sensitive attributes are not included in training data and fairness constraints are not employed. The main contributions are theoretical. The theory shows that implementing a distribution shift during pre-processing improves model generalization on average as well as fairness. Building on previous work employing influence functions, the paper suggests sampling influential examples during training.

**Strengths:**

(1) The idea that better generalization error might also improves fairness performance is an important area of theoretical fairness. There are recent and upcoming papers in a similar vein in applied ML but not as many theory papers

(2) Theorem 3.2 on the upper bound of "fairness disparity" is mathematically interesting and seems to be the main result

**Weaknesses:**

(1) Implementing distribution shift as a way to improve generalization is already known in studies of adversarial training and reliable deep learning, but the paper does not engage with this existing literature

(2) Exposition needs improvement throughout for clarity. For instance, the paper states "Our theoretical analysis indicates that training on datasets with a strategically implemented distribution shift can effectively reduce both the upper bound for fairness disparity and model generalization error (Lemma 3.1, Theorem 3.2). This gives us a key insight that fairness and accuracy can be improved simultaneously even with simply traditional training. [Section 3]" How is training on datasets with a strategic distribution shift simple traditional training? And for the second bullet in contributions, the paper states "we sample influential examples" but does not clarify what influence function is used throughout the paper. I see section 4.1.2 but it is not clear to me how this is distinct from the previous definitions in the literature.

(3) The paper over-emphasizes that sensitive attributes are not used during training, which is pretty standard practice in fair classification. Sure, maybe for post-processing techniques but this particular paper is presenting a technique for pre-processing/during training adjustments. Further, even in work on leave-one-out fairness or influence functions and fairness (analysis of perturbing training data in some way), sensitive attributes are not typically used in training. I notice that these are also areas of related work that the paper does not engage with.

(4) Related work section needs improvement. The section splits into pre-processing methods and post-processing methods which is not helpful for the main contribution of the paper

**Questions:**

1. What is the relationship between the strategic distribution shifts and adversarial training?

2. What is the novelty in section 4? Which parts are restating results from the literature and what parts are new?

3. What is the motivation for employing the distribution shift for this reason? Why is a particular type of shift used? And again, how is this different from adversarial or influence function informed training?

minor:
4. Why are there no captions in the tables?

---

> ### Author Response · Authors · 2023-11-17
> **Part 1: Rebuttal by Authors**
>
> **Response to W1**: We recognize the parallels between our approach and the existing studies in adversarial training and reliable deep learning where distribution shifts are used to enhance generalization. However, this paper specifically targets using distribution shifts to mitigate fairness disparities without harming the generalization. Then, adversarial training is primarily concerned with improving a model's robustness against adversarial attacks. The focus is on ensuring model stability and performance in the face of intentionally crafted input perturbations. Besides, reliable deep learning focuses on ensuring that these models perform consistently under varying conditions and are resilient to errors, noise, and adversarial attacks. Therefore, this work is distinct in that it specifically aims to tackle fairness issues using distribution shifts, a focus not traditionally central to adversarial training or the broader scope of reliable deep learning.
>
> **Response to W2**:  Our training method is akin to conventional training, involving two stages: a warmup process and an incremental process. Initially, we developed a standard machine learning model without any modifications. In the incremental stage, we implement strategic distribution shifts by assessing the influence of samples on prediction and fairness. Samples that exert a negative influence on both prediction and fairness loss — indicating an enhancement in both fairness and prediction — are then solicited for further training.
> Apologies for any confusion regarding the term “influential”. It's analogous to the influence function concept, where we select unlabeled samples based on their influence on prediction (Lemma 4.1) and fairness (Lemma 4.2), hence the term 'influential examples'. More specifically, in Line 6 of the FIS Algorithm, we select those unlabeled samples with negative influences on prediction and fairness. By doing so, we aim to reduce fairness disparity without compromising accuracy.
> Our use of "influential" differs from previous uses in literature, which often focus on identifying misclassified examples from training sets by using influence functions [1,2]. In contrast, we only assess the influence of prediction and fairness on the validation sets $Q_v$ for unlabeled samples by applying first-order gradient estimation, following the traditional training process.
>
> **Response to W3**:  We respectfully disagree with the comment that our paper places undue emphasis on a particular aspect. Previous research often presupposes the availability of demographic details like gender and race, known as sensitive attributes, for training fair models, particularly in in-processing methods.
>
> When sensitive attributes are missing, some works tend to find some proxy models to generate proxy attributes [3]. However, in our method, we need neither true sensitive attributes nor their proxies in the training. There are some others mentioned by **Reviewer B8Mi** and **Reviewer wM5N**, including DRO to improve fairness without sensitive attributes.  Although they also evaluate the worst-group performance in the context of fairness, their approach differs as they do not strive to equalize the loss across all groups.
> In our revised manuscript, we intend to elaborate on other studies that conduct training without using sensitive attributes. Additionally, we respectfully appreciate any relevant references you may provide.
>
> **Response to W4**: Taking into account the feedback from other reviewers, we plan to provide a more comprehensive overview in the 'Related Work' section. This will encompass topics such as distributionally robust optimization, the accuracy-fairness tradeoff, distribution shifts, and adversarial training.
>
> **Response to Q1**: Adversarial training works on existing training data while predicting adversarial behaviors at deployment/test time, while ours (strategic distribution shifts) actively solicits new samples to augment the training dataset.
>
> **Response to Q2**: Motivated by the key observation of theoretical findings, in section 4, we propose an influence-guided sampling strategy that actively samples new data to supplement the training set such as mitigating fairness disparity. Besides, we are confident that the overall content in Section 4 is new. In subsection 4.1.1, we restate the foundational scenarios and further introduce the extra validation set $Q_v$ and the unlabeled set U. Following this, subsection 4.1.2 delves into the methodologies used to determine the influence on predictions and the fairness component. These concepts are integral to the FIS algorithm we describe in Section 4.2.
>
>
> [1] Just train twice: Improving group robustness without training group information, ICML 2021.
>
> [2] Achieving Fairness at No Utility Cost via Data Reweighing with Influence, ICML 2022.
>
> [3] Weak Proxies are Sufficient and Preferable for Fairness with Missing Sensitive Attributes, ICML 2023.

---

> ### Author Response · Authors · 2023-11-17
> **Part 2: Rebuttal by Authors**
>
> **Response to Q3**: The use of distribution shift is inspired by theoretical insights from Lemma 3.1 and Theorem 3.2. These findings shed light on the reasons behind the tradeoff between accuracy and fairness, guiding us to manage this tradeoff by controlling the distribution shift's negative impact on accuracy via generalization error. Without specific information on the train/test distribution, determining the precise distribution shift required is challenging. However, our primary goal is to leverage this distribution shift to minimize the group gap without harming model accuracy.
>
> Our approach differs in its fundamental focus and methodology. While adversarial training is geared towards enhancing a model's robustness against intentionally crafted perturbations, and influence function-informed training aims to understand and mitigate the impact of specific training examples, our method primarily addresses fairness against potential data shifts by adding more examples. We focus on generalization across varying data distributions rather than explicitly countering adversarial examples or pinpointing influential data points.
>
> **Response to Q4**: Sorry for missing this part of the information. We will add captions for tables in the revised version.

---

> > ### Comment · Reviewer_6fEn · 2023-11-21
> > **Thank you for the responses**
> >
> > Thank you to the authors for your comments and responses.

---

> > > ### Author Response · Authors · 2023-11-21
> > >
> > > We sincerely thank the reviewer for the valuable suggestion, and we have incorporated all the discussion and experiment results from the rebuttal into our paper. We hope the reviewer finds our response and revision to the manuscript satisfactory. If the reviewer has any additional suggestions or comments, we are more than happy to address them and further revise our manuscript!

---

### Official Review · Reviewer_B8Mi · 2023-11-01

**Soundness:** 2 fair
**Presentation:** 3 good
**Contribution:** 3 good
**Rating:** 5
**Confidence:** 2

**Summary:**

The paper focuses on addressing the challenge of training fair classifiers with limited number of labeled training samples (without access to sensitive attributes) and an unlabelled dataset. Instead of relying on in-processing fairness constraints due to the lack of sensitive attributes and to avoid accuracy-fairness tradeoff, the authors make a theoretical observation that traditional training on dataset with carefully induced distribution shift can decrease the upper bound of fairness disparity and model error. Based on this observation, the authors propose Fair Influential Sampling (FIS) algorithm. FIS leverages the unlabeled dataset to query examples that help in shifting the training data distribution to maximize fairness without compromising accuracy. The effectiveness of FIS is demonstrated empirically using four datasets.

**Strengths:**

- The problem setting is relevant and interesting.
- The paper addresses the issue of mitigating unfairness without access to sensitive information on the training dataset.

**Weaknesses:**

- There is a large body of work to improve fairness without sensitive attributes on training data [1, 2] and also on validation data [3, 4, 5]. Is there a reason why the authors did not compare the performance of FIS against such methods?
- The related works section lacks discussion about fair active learning frameworks like [6]. In addition, the authors should distinguish the proposed framework from existing fair active learning frameworks and perform an empirical comparison.
- The paper is missing a comparison with in-processing fairness training algorithms, such as MinDiff [7]. Such a comparison would provide a clearer perspective on any accuracy-fairness trade-offs advantages that FIS may entail.
- How are the hyper-parameters, such as number of new examples in each round r, number of rounds T, tolerance $\epsilon$, etc., of the training algorithm determined? This question arises because the selection of hyper-parameters appears to be a significant challenge when training fair models, as reported in [2, 4].
- To gain a comprehensive understanding into the effectiveness of the proposed algorithm, it is imperative to conduct a sensitivity analysis that explores the relationship between the labeling budget and the (accuracy, fairness).

[1] Kirichenko, Polina, Pavel Izmailov, and Andrew Gordon Wilson. "Last Layer Re-Training is Sufficient for Robustness to Spurious Correlations." International Conference on Machine Learning. 2022.
[2] Liu, Evan Z., et al. "Just train twice: Improving group robustness without training group information." International Conference on Machine Learning. PMLR, 2021.
[3] Lahoti, Preethi, et al. "Fairness without demographics through adversarially reweighted learning." Advances in neural information processing systems 33 (2020): 728-740.
[4] Veldanda, Akshaj Kumar, et al. "Hyper-parameter Tuning for Fair Classification without Sensitive Attribute Access." arXiv preprint arXiv:2302.01385 (2023).
[5] Sohoni, Nimit, et al. "No subclass left behind: Fine-grained robustness in coarse-grained classification problems." Advances in Neural Information Processing Systems 33 (2020): 19339-19352.
[6] Anahideh, Hadis, Abolfazl Asudeh, and Saravanan Thirumuruganathan. "Fair active learning." Expert Systems with Applications 199 (2022): 116981.
[7] Prost, Flavien, et al. "Toward a better trade-off between performance and fairness with kernel-based distribution matching." arXiv preprint arXiv:1910.11779 (2019).

**Questions:**

Please see above

---

> ### Author Response · Authors · 2023-11-17
> **Rebuttal by Authors**
>
> **Response to W1**: Work [1] primarily concentrates on improving the worst-group accuracy of neural networks in the presence of spurious features, which seems irrelevant to the fairness topic. Work [2] mainly focuses on the use of distributional robust optimization (DRO) to upweight the training set to improve group robustness. Similar to DRO [2], [3] proposes an optimization approach, Adversarially Reweighted Learning, to improve worst-case performance over unobserved protected groups. In reality, [4] is an expansion of the work presented in [2], with a primary focus on the fine-tuning of hyper-parameters. This extension specifically addresses the sensitivity of various hyper-parameters, such as the size of the validation set. [5] exploits these estimated subclasses by training a new model to optimize worst-case performance over all estimated subclasses using group distributionally robust optimization (GDRO).
> These works explicitly focus on improving the worst-group loss. Although they also evaluate the worst-group performance in the context of fairness, their approach differs as they do not strive to equalize the loss across all groups. Besides, in these studies, accuracy and worst-case accuracy are used to showcase the efficacy of DRO. Essentially, they equate fairness with uniform accuracy across groups, implying a model is considered fair if it demonstrates equal accuracies for all groups. However, this specific definition of fairness is somewhat restrictive and does not align with more conventional fairness concepts like Demographic Parity (DP) or Equalized Odds (EOD).
>
> **Response to W2**: While fair active learning frameworks may seem similar to our approach, there are crucial distinctions. For instance, work [6] focuses on sampling additional unlabeled data points based on a multi-objective optimization for misclassification error (Shannon entropy) and fairness regularization (demographics parity). In contrast, [6] still requires knowledge of the sensitive attributes of unlabeled samples, while our method does not necessitate such information. Practically, without this information, it's difficult to assess fairness disparity (e.g., using demographics parity) and determine if a model is fair. To overcome this, we resort to a validation set that includes sensitive attributes, providing a reliable measure of fairness disparity.
>
> **Response to W3**: Those in-processing fairness training algorithms typically require sensitive attribute information to develop regularization terms. For instance, MinDiff [7] employs various regularization methods to reduce bias so as that obtain an accuracy-fairness tradeoff. Consequently, comparing our approach with in-processing algorithms is unnecessary and not fair, as these algorithms are not applicable in scenarios where sensitive attribute information in the training data is unavailable.
>
> **Response to W4**: The fine-tuning of hyper-parameters, such as the size of the validation set, is crucial due to the use of proxies and the implementation of Distributionally Robust Optimization (DRO) [2, 4]. More specifically, [2] employs misclassified examples from a standard Empirical Risk Minimization (ERM) model to represent minority sub-groups, while [4] generates pseudo-sensitive attribute labels on validation data. In contrast, our approach utilizes a straightforward, traditional gradient-based sampling method, simplifying implementation and reducing the complexity of hyper-parameter fine-tuning. In our experiments, it is important to note that the hyperparameters specific to each dataset are the same regardless of the sensitive attributes or the binary classification targets.
> In our experimental setup, it's crucial to point out that the hyperparameters specific to each dataset remain constant.
> To address your concern, we will conduct experiments on these hyper-parameters, including the impact of solicitation budgets. These additional results will be added to the revision before the rebuttal deadline.
>
> **Response to W5**: Thank you for your feedback. Regarding your concern, we plan to conduct a sensitivity analysis to examine how the labeling budget impacts both accuracy and fairness.

---

> > ### Author Response · Authors · 2023-11-19
> > **Additional experiments for Weakness 5**
> >
> > Thank you for your patience. We have completed the additional experiments as previously discussed, examining how the labeling budget impacts both accuracy and fairness. We present the results of test accuracy and fairness
> > disparity across different label budgets on the CelebA, Compas, and Jigsaw datasets. In these experiments, we
> > use the demographics parity (DP) as our fairness metric. For convenience, we maintain a fixed label budget
> > per round, using rounds of label budget allocation to demonstrate its impact.  The designated label budgets
> > per round for the CelebA, Compas, and Jigsaw are 256, 128, and 512, respectively. In the following figures,
> > the x-axis is both the number of label budget rounds. The y-axis for the left and right sub-figures are test
> > accuracy and DP gap, respectively. The corresponding results have been shown in Figures 1–3. We can observe that, compared to the BALD baseline, our approach substantially reduces the DP gap without sacrificing test accuracy.

---

> > > ### Author Response · Authors · 2023-11-23
> > >
> > > We sincerely thank the reviewer for the valuable suggestion, and we have incorporated all the discussion and experiment results from the rebuttal into our paper. We hope the reviewer finds our response and revision to the manuscript satisfactory. If the reviewer has any additional suggestions or comments, we are more than happy to address them and further revise our manuscript!

---

### Official Review · Reviewer_UQLS · 2023-11-01

**Soundness:** 3 good
**Presentation:** 3 good
**Contribution:** 2 fair
**Rating:** 5
**Confidence:** 5

**Summary:**

This paper studies the fairness problem with a sampling strategy. From the motivation that changing the data distribution would help with the fairness issue, the authors propose to estimation the influence of sample via first-order gradient approach, and re-weight samples individually. Experiments are conducted on images, tabular data, and language to demonstrate their approach.

**Strengths:**

1. The proposed algorithm is technically sound and straightforward. No access to the sensitive attribute of training data is a good property to have.

2. The presentation is clear and easy to follow.

3. The experimental datasets cover multiple types of data, which is good to have.

**Weaknesses:**

1. Key reference missing. In [1], the authors also proposed a sampling/reweighing strategy to select good samples based on influence estimation. The algorithm has no access to the sensitive attributes of training data. The two paradigms seem conceptually and technically similar to me. Some discussions are needed in this paper.

2. Weak connection between the theorem and algorithm. I understand the theorem served as a very high-level motivation to change the distribution shift, but from my perspective, it has limited connections to the specific algorithm later on. Also the theorem is closely related to the theorems in domain adaptation established several years ago, so that somehow to be incremental.

3. Since the influence is based on first-order estimation, if there any chance to validate the estimation of influence per sample? Maybe show the actual influence and its estimation would be helpful.

4. The authors consider the accuracy in the algorithm. However, in the experimental section, I didn't see any numerical evaluation for model accuracy. Given the context, involving the model's accuracy and show the tradeoffs between fairness and accuracy make more sense to me.

[1] Achieving Fairness at No Utility Cost via Data Reweighing with Influence, ICML'22

**Questions:**

N/A

---

> ### Author Response · Authors · 2023-11-17
> **Rebuttal by Authors**
>
> **Response to W1**: We appreciate your observation and agree that there are similarities in methods, yet our approach significantly diverges in its specifics. First, unlike the influence function method, we employ a first-order gradient to select the samples, verified in Lemma 4.1 and Lemma 4.2.  We would like to emphasize that our method focuses on soliciting additional samples from an external unlabeled dataset (consistent with a fair active learning framework) while [1] reweights the existing and fixed training dataset. Benefiting from this, our approach is more suitable for larger numbers of unlabeled data because it allows for straightforward incremental training on a basic model, rather than the complete retraining required by [1].
>
> **Response to W2**: Due to the lack of specific information about the train/test distribution and sensitive attributes in our scenario, crafting a perfectly tailored algorithm that directly links to our theorems to control distribution shifts is challenging. To address this, we've introduced the heuristic sampling method FIS, focusing on utilizing gradient descent to effectuate an appropriate distribution shift. While our work aligns with certain domain adaptation theories, it primarily investigates the underlying causes of the accuracy-fairness tradeoff, extending beyond just domain adaptation. By controlling the adverse effects of distribution shifts, our goal is to strike a more favorable tradeoff between accuracy and fairness.
>
> **Response to W3**: The choice of first-order estimation, rather than considering the second-order information, is due to the nature of training deep neural networks with non-convex objectives subject to fairness constraints.  In such scenarios, the model often does not converge to a local minimum where the first-order gradient can be treated as zero, thus the first-order term will dominate over the second-order term. In recent literature [R1], the authors have shown that the first-order estimation per sample is nearly accurate under fairness constraints.
> In the uploaded revision, we added Figure 5 to compare the estimated influence with the actual influence. We observe that while some of the examples are away from the diagonal line (which indicates the estimation is inaccurate), the estimated influence for most of the data samples are very close to their actual influence values.
>
> **Response to W4**: Apologies for not providing sufficient information in the table captions. We will revise the table captions to include more details and upload the revised version. In those tables, we report the values of accuracy and corresponding three fairness metrics in the format: **(test accuracy, fairness metric)**. Although the tradeoff might not always be apparent in practice, the results shown in the tables demonstrate that we can reduce fairness disparity without sacrificing model accuracy.
>
> [R1] Understanding instance-level impact of fairness constraints, ICML 2022.

---

> > ### Comment · Reviewer_UQLS · 2023-11-19
> > **Thanks for your response**
> >
> > Thanks for your valuable time and effort, I really appreciate it!
> >
> > W1: I agree with you. These two methods are technically different. However, they do share something in terms of sample-level modeling. Proper discussions can better guide readers to more work in sample-level modeling for fairness.
> >
> > W2:  I totally understand the challenge between theorem and practice, and thanks for acknowledging it.
> >
> > W3: I might be wrong, but I didn't see a figure in the current pdf. Did you upload the most recent paper?
> >
> > W4: Agree, better captions are needed. Make sure the readers can find all relevant information about the tables at their first glance.

---

> ### Author Response · Authors · 2023-11-19
> **Rebuttal by Authors**
>
> Thanks for your comment! We apologize for any inconvenience. We have uploaded the latest paper. In revision, more discussion has been added in Section 2 (Related Work). Then, Figure 4 compares the estimated impact with the actual impact. Please always feel free to let us know if we could assist with understanding the paper or addressing any concerns.

---

> > ### Comment · Reviewer_UQLS · 2023-11-19
> > **Thanks for your response**
> >
> > Yes, this time works, I can see the updated paper. Thanks for your effort.
> >
> > Regarding the table and caption, I would suggest putting all relevant information about tables into its caption. Maybe not putting the tables and how to read the table in two different rooms. Readers may not go through all the context before the tables carefully, so a caption can help in this case.
> >
> > Although effort has been made, I still have some concerns regarding the technical novelty and contributions of this paper. Also, the gap between theory and practice diminishes the value of contributions.
> >
> > To acknowledge the effort paid by the authors and the clarifications, I would like to raise my score to 5.

---

> > > ### Author Response · Authors · 2023-11-21
> > > **Thanks for your comments**
> > >
> > > Thank you for the thoughtful comments and considering raising your score! For convenience, we will consider adding all relevant information about tables into its captions. Here, we would like to shed more light on the technical novelty and contribution of this paper. The key point is that the developed sampling method FIS does not rely on any sensitive attributes or their proxies in train data. Indeed, mitigating fairness disparity without access to sensitive attribute information is challenging, as we cannot assess whether the model is fair without using fairness measures constructed from sensitive attribute information.
> > > As noted by Reviewers B8Mi and wM5N, other lines of work use Distributionally Robust Optimization (DRO) to mitigate fairness disparity in scenarios lacking sensitive attributes. However, as we mentioned in the related work section, these studies primarily focus on improving the worst-group loss, rather than equalizing loss across all groups. Additionally, these studies use accuracy and worst-case accuracy to demonstrate the efficacy of DRO. Essentially, they equate fairness with uniform accuracy across groups, implying that a model is considered fair if it demonstrates equal accuracies for all groups. This definition of fairness is somewhat restrictive and does not align with more conventional fairness definitions like Demographic Parity (DP) or Equalized Odds (EOD). These works have only managed to improve the worst-group loss within a specific fairness definition. This limitation underscores the broader applicability and benefits of our approach.
> > >
> > > Regarding theoretical concerns, our standpoint is to avoid making further assumptions or accessing additional information to mitigate fairness disparities, which is more practical. With more reasonable assumptions (e.g., distribution shifts between train and test datasets), it is possible and reasonable to develop algorithms that more effectively mitigate fairness disparities. Therefore, we believe that our method provides more insights to address the significant challenge in scenarios where sensitive attributes (i.e., protected or expensive group annotations) are unavailable.

---

### Official Review · Reviewer_wM5N · 2023-11-03

**Soundness:** 3 good
**Presentation:** 3 good
**Contribution:** 2 fair
**Rating:** 6
**Confidence:** 4

**Summary:**

The paper proposes a new fair training method when the sensitive attributes are missing. The paper first provides theoretical analyses to show the upper bounds for the generalization error and fairness disparity of the model. For example, their theoretical observations show that the upper bound of the fairness disparity is affected by both distribution shifts and group bias in the data. Based on such analyses, the paper proposes a new sampling strategy that utilizes the influence information of each unlabeled sample to improve the fairness and accuracy of the model. The proposed algorithm is tested on several datasets, including image and tabular datasets.

**Strengths:**

- The paper focuses on a realistic setting in model fairness, where the sensitive attribute labels are unavailable during the training.
- The paper provides interesting theoretical analyses, including the upper bounds of generalization error and fairness disparity.
- The paper uses various benchmark datasets, including both image and tabular scenarios, helping to show the multiple applications of the proposed algorithm.

**Weaknesses:**

- The paper needs to clarify the connection between this paper and other related works in the fairness literature.
  - For example, one of the important discussions in the paper is about the accuracy-fairness tradeoff, which also affects the main proposed algorithm. However, such a tradeoff between accuracy and fairness has been widely studied (e.g., [1]), so it would be better if the paper could clarify what is the difference between this paper’s analysis and previous discussions in the fairness literature.
  - Also, the paper uses the concept of distribution shifts, which is recently extensively studied in the fairness literature, but the paper does not discuss those works (e.g., [2, 3, 4]). It seems the setups of this paper and recent distribution shifts studies in fairness literature are a bit different, as this paper aims to ‘utilize’ an appropriate distribution shift for sampling, while many recent studies focus on ‘solving’ the distribution shifts between training and target distributions for fair training. Thus, it would be much better if the paper could clarify the connection and differences between this work and other distribution shifts studies.
- The paper does not compare with enough baselines in their experiments. For example, there are various algorithms for training fair models without using sensitive attribute information (e.g., [5, 6]), but the paper does not include clear comparisons with those works.

-------------
[1] Menon and Williamson, The cost of fairness in binary classification, FAT* 2018.

[2] Singh et al., Fairness Violations and Mitigation under Covariate Shift, FAccT 2021.

[3] Roh et al., Improving Fair Training under Correlation Shifts, ICML 2023.

[4] Giguere et al., Fairness Guarantees under Demographic Shift, ICLR 2022.

[5] Hashimoto et al., Fairness without demographics in repeated loss minimization, ICML 2018.

[6] Lahoti et al., Fairness without demographics through adversarially reweighted learning, NeurIPS 2020.

**Questions:**

The key questions are included in the above weakness section.

-------------------
[After rebuttal] I read both the responses and the revised paper. As most of my concerns have been resolved, I raised my score.

---

> ### Author Response · Authors · 2023-11-17
> **Rebuttal by Authors**
>
> **Response to W1-1**: We would like to clarify that we do not work in the classical regime of fairness-accuracy tradeoff. By properly collecting new data from an unlabeled dataset, we can improve both generalization and fairness at the same time which can not be achieved by working on a fixed and static training dataset that naturally incurs such tradeoff.
> Another notable distinction between our work and previous studies on the accuracy-fairness tradeoff lies in the handling of sensitive attributes of training data. For instance, [1] achieves this tradeoff using fairness regularization (in-processing techniques), which necessitates sensitive attribute information. In contrast, our paper's key contribution is the elimination of this information, thus offering a more practical and applicable framework in scenarios where such information is unavailable.
>
>
> **Response to W1-2**: Thanks for the suggestions. Overall, our approach significantly differs from other studies that concentrate on fairness disparity arising from train-test distribution shifts. Commonly, such research necessitates extra assumptions to build theoretical connections between features and attributes, like causal graphs [2], correlation shifts [3], and demographic shifts [4]. In contrast, our approach refrains from making further assumptions about the characteristics of distribution shifts. Instead, we directly utilize sampling methods to construct an appropriate distribution shift, aiming to achieve a balanced tradeoff between accuracy and fairness.
> More specifically, work [2] uses complex causal graphs to hypothesize about distribution shifts between training and testing datasets. Following this, Then, [3] introduces and analyzes the correlation shifts between train and test distribution, serving to boost existing fair in-processing methods. However, to our knowledge, fair in-processing methods necessarily require information on sensitive attributes during training, whereas our approach does not require this information. [4] control unfairness of a model by making the demographic shifts assumptions, where the marginal distribution of the data remains the same conditional on the subgroup but the subgroup distribution can change.
>
> **Response to W2**:   [5] proposes an approach based on distributionally robust optimization (DRO), which minimizes the worst-case risk over all distributions close to the empirical distribution. Then, [6] also proposes an optimization approach, Adversarially Reweighted Learning, to improve worst-case performance over unobserved protected groups, indicating the main goal is to improve the worst-group loss. Although these two works [5, 6] also evaluate the worst-group performance in the context of fairness, their approach differs as they do not strive to equalize the loss across all groups. Besides, in these studies, accuracy and worst-case accuracy are used to showcase the efficacy of DRO. Essentially, they equate fairness with uniform accuracy across groups, implying a model is considered fair if it demonstrates equal accuracies for all groups. However, this specific definition of fairness is somewhat restrictive and does not align with more conventional fairness concepts like Demographic Parity (DP) or Equalized Odds (EOD).
> In response to your concern, we are considering using one of DROs (e.g., [5]) as a baseline for our ongoing experiments. These additional results will be added to the revision before the rebuttal deadline.

---

> > ### Author Response · Authors · 2023-11-21
> > **additional experiments for weakness 2**
> >
> > Thank you for your patience. In response to your concern, we have added a baseline JTT-20 [R1] in our revised manuscript. This baseline specifically reweights misclassified examples, assigning them a weight of 20 during the retraining process. We have updated the results for the four datasets in the corresponding tables to reflect this addition. These updates also highlight the effectiveness of our proposed method.
> >
> > [R1] Liu, Evan Z., et al. "Just train twice: Improving group robustness without training group information." International Conference on Machine Learning. PMLR, 2021.

---

> > > ### Author Response · Authors · 2023-11-23
> > >
> > > We have incorporated all the discussion and experiment results from the rebuttal into our paper. We hope the reviewer finds our response and revision to the manuscript satisfactory. If the reviewer has any additional suggestions or comments, we are more than happy to address them and further revise our manuscript!

---

> > > > ### Comment · Reviewer_wM5N · 2023-11-23
> > > > **Thank you for the response**
> > > >
> > > > I appreciate the authors' efforts to address my concerns. I read both the responses and the revised paper.
> > > > As most of my concerns have been resolved, I raised my score.
> > > >
> > > > I would like to suggest the authors further clarify the accuracy-fairness tradeoff-related discussion in the paper. The accuracy-fairness tradeoff theories in previous works can also be applied when we do not have group labels during training. Such a tradeoff is an inherent one; thus, one can think that using the group information during training will not affect such an inherent tradeoff.

---

### Official Review · Reviewer_FJTt · 2023-11-08

**Soundness:** 2 fair
**Presentation:** 2 fair
**Contribution:** 3 good
**Rating:** 5
**Confidence:** 2

**Summary:**

This paper studies learning fair classifiers without implementing fair training algorithms to avoid possible leakage of sensitive information. Its analysis indicates that training on datasets with a strategically implemented distribution shift can effectively reduce both the upper bound for fairness disparity and model generalization error. It also proposes the sampling algorithm FIS to sample influential examples from an unlabeled dataset.

**Strengths:**

1. Sampling influential examples from an unlabeled dataset based on the combined influences of prediction and fairness is interesting.
2. The theoretical analysis on the upper bound of the fairness disparity is provided.
3. The experiments on three datasets demonstrate the proposed algorithm is useful.

**Weaknesses:**

1.	The definition 3.1's symbolism is not clear, are $P$ and $Q$ the same as preliminaries? Why use the model trained on P instead of that trained on Q? Could you give more explanation?
2.	Assumption 3.2 seems a bit strong. The assumption before Lemma 3.1 that the loss is bounded is not common. Could you give more justification to these assumptions?
3.	In the first paragraph in Sec 4.1.1, the assumption of an auxiliary hold-out validation dataset is too strong. For my understanding, test data means that we don’t know the distribution of the data. So I am not sure the reasonability of the assumption.
4.	Although it states the computation cost of the proposed algorithm is low, it seems the algorithm needs to pre-calculate the loss for testing the performance of a sample, which is costly.
5.	It lacks discussion of how to select the initial training dataset for the warm start (influence when applying different proportions or distributions), and how to determine the solicitation budget $r$ which is the declared a small number of sampling data to gain a better result (both accuracy and fairness).
6.	Regarding the experiments, the baselines are not sufficient.

Minors:
1.	The symbol used in paper should be unified. Notion of Q and P are not used consistently in Sec 3 and 4.
2.	In the proposed algorithm section, the proposed strategy I or II should be in Line 6, and the calculation of prediction's and fairness's influences should be in Line 7, 8.
3.	Typo: That -> that in paragraph before Def. 3.2

**Questions:**

1.	It is not clear about the statement "an appropriate distribution shift can reduce both the upper bound for fairness disparity and model generalization error". I think the Theorem 3.2 tells us that no distribution shifts will help lead to the smaller generalization error bound, and a smaller shift leads to smaller error (straightforward).
2.	In Eq. 1, if $f$ is a classifier, then $x$ seems to be the feature instead of original data.

---

> ### Author Response · Authors · 2023-11-17
> **Part 1: Rebuttal by Authors**
>
> **Response to W1**: We apologize for any raised confusion. In our work, we represent the training dataset using the notation P and the test dataset with the notation Q, without assuming specific characteristics for these sets. This implies that P and Q could either be independent and identically distributed (iid) or exhibit a distribution shift. Conventionally,  it's a standard approach to train a model on the training set P and evaluate its effectiveness on the test set Q.
>
> **Response to W2**: We would like to clarify that the assumption of L-lipschitz smoothness (Assumption 3.1) and bounded gradients  (Assumption 3.2) of the empirical risk are common and basic in the theoretical analysis of stochastic gradient descent (SGD) and related optimization methods in machine learning [1, 2, 3]. For your reference, these two assumptions are similar to Assumption 1 and Assumption 4 presented in [1], respectively. More specifically, the L-Lipschitz property for the empirical risk is inferred from the L-Lipschitz property of the loss function because the empirical risk $R_P(w)$ is defined as the average of the loss function over the training dataset. On the other hand, gradient clipping is a widely applied trick to avoid exploding gradients in deep learning, thus implying that the bounded gradient is a decent assumption.
>
>
> **Response to W3**: We respectfully disagree with the comment that the assumption of the validation dataset is too strong. This assumption is widely recognized and has been presented in numerous existing works [4-8]. In our setting, the distribution of the data remains unknown to us because merely accessing a small validation dataset for testing purposes does not fully reveal the data distribution.
> Besides, we can analyze this assumption in two scenarios: whether the train and test distributions are independent and identically distributed (iid) or not. If they are i.i.d., the validation dataset can be drawn from the training dataset. However, in cases of distribution shifts between train and test data, we cannot ascertain the influence of solicited data samples without any explicit knowledge of the test data. Therefore, we utilize this reasonable assumption to gain some insight into the test dataset or its distribution.
>
>
> **Response to W4**: We would like to clarify that the extra computation cost is comparable to the cost of traditional model training. The main extra computation cost in FIS (Algorithm 1) is the cost of model gradients. Let $p$ denote the number of model parameters, then the cost for computing the gradients is $O(p)$ per sample. Specifically, in each round that we need sampling, we need to calculate three parts of gradients: the gradients of $N_U$ unlabeled instances, the average gradient of $N_v$ validation instances wrt accuracy loss, and the gradient of $N_v$ validation instances wrt fairness loss. Note in general, $N_v \ll N_U$. In practical implementation, to speed up the calculation of gradients over $N_U$ instances, we randomly sample 0.2～0.5 percent of the unlabeled dataset in each sampling batch. Additionally, we can increase $r$ to save computation costs. In our experiments, we usually have 10～20 sampling rounds. On a single GPU, running one experiment for CelebA requires about 4 hours.
>
> **Response to W5**:  In our work, we do not make specific assumptions on the initial training dataset. In our experiments, we randomly select a subset of examples as the initial labeled training set. In response to your concern, we are conducting experiments to show the impact of solicitation budgets. These additional results will be added to the revision before the rebuttal deadline.
>
>
> [1] On the Convergence of FedAvg on Non-IID Data, ICLR 2020.
>
> [2] Fair resource allocation in federated learning, ICLR 2019
>
> [3] SCAFFOLD: Stochastic Controlled Averaging for Federated Learning, ICML 2020.

---

> > ### Author Response · Authors · 2023-11-19
> > **Additional experiments for weakness 5**
> >
> > Thank you for your patience. We have completed the additional experiments as previously discussed, examining how the labeling budget impacts both accuracy and fairness. We present the results of test accuracy and fairness disparity across different label budgets on the CelebA, Compas, and Jigsaw datasets. In these experiments, we use the demographics parity (DP) as our fairness metric. For convenience, we maintain a fixed label budget per round, using rounds of label budget allocation to demonstrate its impact. The designated label budgets per round for the CelebA, Compas, and Jigsaw are 256, 128, and 512, respectively. In the following figures, the x-axis is both the number of label budget rounds. The y-axis for the left and right sub-figures are test accuracy and DP gap, respectively. The corresponding results have been shown in Figures 1–3. We can observe that, compared to the BALD baseline, our approach substantially reduces the DP gap without sacrificing test accuracy.

---

> ### Author Response · Authors · 2023-11-17
> **Part 2: Rebuttal by Authors**
>
> **Response to W6**: As far as we are aware, there don't seem to be fitting baselines for comparison in our case. We are receptive to any suitable baseline suggestions you might have. Although some studies train models without using sensitive attributes [4-8], they may not serve as direct baselines for our work. This is because these works explicitly focus on improving the worst-group loss. Although they also evaluate the worst-group performance in the context of fairness, their approach differs as they do not strive to equalize the loss across all groups. Besides, in these studies, accuracy and worst-case accuracy are used to showcase the efficacy of DRO. Essentially, they equate fairness with uniform accuracy across groups, implying a model is considered fair if it demonstrates equal accuracies for all groups. However, this specific definition of fairness is somewhat restrictive and does not align with more conventional fairness concepts like Demographic Parity (DP) or Equalized Odds (EOD).
> In addressing your concern, we plan to use one of DROs [4, 7, 8] as a baseline in our current experiments. We will incorporate these results into the revised manuscript before the deadline for rebuttals.
>
> **Response to Weakness (minors)**: Sorry for these typos. We will carefully revise the references in the revision.
>
> **Response to Question 1**: This seems to be a straightforward understanding that theorem 3.2 tells you that if there is no distribution shift between the training and testing set, the generalization error would be small and bounded. While avoiding distribution shifts contributes to a small error, this also maintains a large fairness disparity, evident as a group gap in Theorem 3.2. This observation has been validated by numerous existing works that focus on the accuracy-fairness tradeoff.
> In this paper, we are trying to build connections between train and test data by using our actively enhanced data (including the initial labeled training data and new solicited data), following a train data $\rightarrow$ solicited data $\rightarrow$ test data sequence. If we can shift the distribution of training data towards that of test data, we can reduce the fairness (group) gap. It's important to note that we cannot control the fairness gap from train to test data, which is determined by the nature of the problem. Our control lies in strategically shifting the training data distribution by adding more samples, with the hope that this will reduce the inherent fairness gap between train data and test data.
>
> **Response to Question 2**: Your understanding is correct. We define $x$ as features in the first paragraph of Section 3.1(Preliminaries).
>
>  [4] Just train twice: Improving group robustness without training group information, ICML 2021.
>
> [5] Achieving Fairness at No Utility Cost via Data Reweighing with Influence, ICML'22
>
>  [6] Fairness without demographics through adversarially reweighted learning, NeurIPS 2020.
>
> [7] Hyper-parameter Tuning for Fair Classification without Sensitive Attribute Access, arXiv 2023.
>
> [8] No subclass left behind: Fine-grained robustness in coarse-grained classification problems, NeurIPS 2020.

---

> > ### Author Response · Authors · 2023-11-21
> > **Additional experiments for weakness 6**
> >
> > Thank you for your patience. We have added a baseline JTT-20 [R1] in the revision. The baseline JTT-20 will reweight those misclassified examples for retraining. Here, we examine a weight of 20 for misclassified examples in JTT. The corresponding results of four datasets have been updated in tables, which also show the effectiveness of our proposed method.
> >
> > [R1] Liu, Evan Z., et al. "Just train twice: Improving group robustness without training group information." International Conference on Machine Learning. PMLR, 2021.

---

> > > ### Author Response · Authors · 2023-11-23
> > >
> > > We thank the reviewer for the useful suggestion, and we have incorporated all the discussion and experiment results from the rebuttal into our paper. We hope the reviewer finds our response and revision to the manuscript satisfactory. If the reviewer has any additional suggestions or comments, we are more than happy to address them and further revise our manuscript!

---

### Meta-Review · Area_Chair_VE5f · 2023-12-04

**Metareview:**

I have read all the materials of this paper including the manuscript, appendix, comments, and response. Based on collected information from all reviewers and my personal judgment, I can make the recommendation on this paper, *reject*. No objection from reviewers who participated in the internal discussion was raised against the reject recommendation.

**Research Question**

The authors consider the active learning for sample annotation to achieve utility and fairness, where the authors consider the scenario that the sensitive attributes of training data are unknown. I have no idea why the authors focus on such a scenario. For the technical part, the authors employ the influence function for sample selection, where the sensitive attributes are not a must but the gradient is the key. In another word, knowing the sensitive attributes or not have little impact on the proposed method.

**Challenge Analysis**

The authors pointed out the core challenges lie in no access to the sensitive attributes or training labels before the actual sampling happened. For the first challenge, I do not think the authors proposed any novel technique to tackle this. But for the second one (no label access), which makes the gradient calculation impossible, I count this one as a true challenge.

**Related Work**

The huge body of active learning is ignored.

**Philosophy**

The authors employed the proxy labels or the low-influence labels, where the first one is a standard baseline in active learning area. In the paper, the authors mentioned that these two strategies will produce similar labels. If this is true, there is no need to calculate the sample influence, which is complicated than the first strategy.

**Theoretical Analysis**

The authors provided some theoretical analysis on the utility-and-fairness tradeoff from the perspective of distribution shift. In the literature, there exist some theoretical analysis on this topic. Some reviewers provided some key references, so I omitted here. But the authors started from the distribution shift, which should be given some credit.

However, one reviewer pointed out the theorems in Section 3 is disconnected with Section 4. The high-level insights does not provide any practical guidance for algorithm design.

**Techniques**

The authors proposed two strategies, the proxy labels or the low-influence labels. However, I am confused why they are similar. Please refer to the Philosophy part.

The core technique in this paper is influence function, which has some limitations for non-convex model. I suggest the authors add some discussion along this direction, which helps understand the suitable conditions of the proposed method.

**Experiments**

1. As one reviewer pointed out, the competitive methods are not enough. The authors ignored the active learning area. Here I provided one extra [1], which also employs influence function for active learning.

2. Standard deviations are suggested for tables.

3. Beyond the current experiments, it is suggested to verify whether the newly added samples have the similar gradient direction with the validation set.

**Presentation**

I strongly suggest the authors take action for presentation and organization. In the current version, every component is related but not coherent to other parts. In this paper, there are many concepts including no sensitive attributes, utility-and-fairness tradeoff, distribution shift, active learning; however, a main roadmap is needed for the selling points.

[1] Influence selection for active learning. ICCV 2021.

**Justification For Why Not Higher Score:**

This paper is not self-standing and does not reach the bar of ICLR.

**Justification For Why Not Lower Score:**

N/A

---

### Decision · Program_Chairs · 2024-01-16

Reject